# RETHINKING GRAPH OUT-OF-DISTRIBUTION GENERALIZATION: A LEARNABLE RANDOM WALK PERSPECTIVE

## ABSTRACT

Out-Of-Distribution (OOD) generalization has gained increasing attentions for machine learning on graphs, as graph neural networks (GNNs) often exhibit performance degradation under distribution shifts. Existing graph OOD methods tend to follow the basic ideas of invariant risk minimization and structural causal models, interpreting the invariant knowledge across datasets under various distribution shifts as graph topology or graph spectrum. However, these interpretations may be inconsistent with real-world scenarios, as neither invariant topology nor spectrum is assured. In this paper, we advocate the learnable random walk (LRW) perspective as the instantiation of invariant knowledge, and propose LRW-OOD to realize graph OOD generalization learning. Instead of employing fixed probability transition matrix (i.e., degree-normalized adjacency matrix), we parameterize the transition matrix with an LRW-sampler and a path encoder. Furthermore, we propose the kernel density estimation (KDE)-based mutual information (MI) loss to generate random walk sequences that adhere to OOD principles. Extensive experiment demonstrates that our model can effectively enhance graph OOD generalization under various types of distribution shifts and yield a significant accuracy improvement of 3.87% over the best of state-of-the-art graph OOD generalization baselines.

## 1 INTRODUCTION

Graph neural networks (GNNs) have become a fundamental solution of encoder architectures for modeling graph-structured data (Wu et al. (2020); Zhou et al. (2022); Bessadok et al. (2022); Song et al. (2022)). They facilitate the efficient computation of node representations, which can be readily adapted to a wide range of graph-based applications, including social network analysis, recommendation systems, anomaly detection and so on (Zhao et al. (2021); Virinchi et al. (2022); Tang et al. (2022); Chen et al. (2022a)). Despite great advances of GNNs, most of existing models follow the i.i.d. assumption, i.e., the testing nodes independently generated from an identical distribution as the training ones (Kipf and Welling (2017); Veličković et al. (2018); Hamilton et al. (2017); Pei et al. (2020); Sun et al. (2023); Li et al. (2024)). However, this assumption doesn't necessarily conform to real-world scenarios since spurious correlations among datasets may infect GNNs' training. Recent evidence has demonstrated that GNNs perform unsatisfactorily on Out-Of-Distribution (OOD) data where the distributions of test data exhibit a major shift compared to the training data (Arjovsky et al. (2019); Koyama and Yamaguchi (2020); Chen et al. (2022b)). Thus, such problem, also known as graph OOD generalization, remains a great challenge to be solved.

Existing graph OOD generalization models for node-level tasks are largely inspired by the concepts from the invariant risk minimization (IRM) and structural causal models (SCMs) (Arjovsky et al. (2019); Koyama and Yamaguchi (2020); Chen et al. (2022b)). These models employ various mechanisms to extract invariant knowledge shared between the training and testing datasets and discard the spurious correlation among them. Broadly, graph OOD generalization models can be categorized into two primary approaches: capturing invariant graph topology and capturing invariant graph spectrum, as shown in Figure 1(b) (Wu et al. (2022); Guo et al. (2024); Xia et al. (2023); Zhu et al. (2021); Liu et al. (2022); Wu et al. (2024); Zhu et al. (2024)). The first approach interprets invariant knowledge as specific graph topology and leverages techniques such as pseudo-environment-generation to facilitate graph OOD generalization learning. In contrast, models that focus on invariant graph spectrum

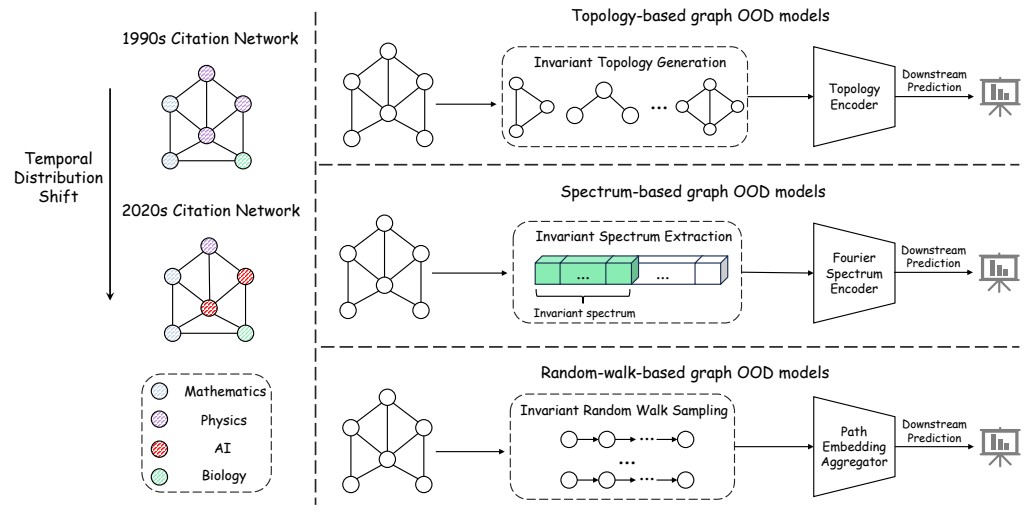

(a) Example of Heterophilic Citation Network    (b) Existing Topology- and Spectrum-based Pipelines & Random-walk-based Pipeline

Figure 1: An example of the heterophilic citation network under temporal distribution shift and pipelines of existing graph OOD models. Topology-based and spectrum-based pipelines are the two primary approaches for graph OOD generalization, while the random-walk-based pipeline is the proposed one in this paper.

learning emphasize the acquisition of a stable spectral representation across graphs under multiple environments. These models typically assume that certain spectral components, particularly the low-frequency spectrum, remain invariant. Thus, they introduce perturbations to the remaining spectral components, thereby generating diverse graph data for graph OOD generalization learning. Despite the advancements in the aforementioned methods, they still fail to achieve satisfactory performance in graph OOD generalization learning. Take the heterophilic citation networks of Figure 1(a) as an example, where nodes represent papers and edges denote citation relationships. The objective in such networks is to predict the category of a given paper (e.g., mathematics, physics, AI and biology), and the latent distribution shifts arise due to temporal variations in citation patterns. For instance, in the 1990s, mathematics-related papers were predominantly cited by works in physics and biology, whereas in the 2020s, mathematics-related papers within a similar local topology may exhibit a stronger tendency to be cited by AI-related research. In this scenario, previous graph OOD generalization models exhibit severe limitations as follow: (**L1**): The presence of an invariant graph topology across graphs is not assured, as structurally similar topologies may correspond to distinct semantic interpretations under varying distribution shifts. For instance, in the citation network depicted in Figure 1(a), the semantic significance of the local topology formed by nodes around physic papers differs from that of nodes around AI papers, despite the fact that both subgraphs exhibit analogous topological structures. (**L2**): The extraction of a universally invariant graph spectrum remains unreliable due to the lack of a well-defined theoretical relationship between the graph spectrum and the formulation of OOD generalization in graphs. For example, in the citation network depicted in Figure 1(a), the graph from the 1990s exhibits stronger homophily, resulting in higher magnitudes in the low-frequency spectrum components and lower magnitudes in the high-frequency components. In contrast, the graph from the 2020s demonstrates increased heterophily, leading to a complete reversal in the distributions across frequency bands. Consequently, no invariant graph spectrum persists in such graphs under temporal distribution shifts.

To address the aforementioned limitations, we utilize learnable random walk sequences as a means of capturing invariant knowledge across graphs under various distribution shifts for node-level tasks. The underlying motivations are derived from two perspectives: (**M1**) Different from the two approaches illustrated above, learnable random walk sequences are capable of integrating graph topology and node features together into the probability matrix, thus concretizing invariant knowledge into the probability of the next random walk. Intuitively, such learnable-probability-based invariant random walk exists as long as there are invariant knowledge shared across graphs under distribution shifts. (**M2**) Through rigorous mathematical analysis, we demonstrate that learnable random walk sequences exhibit a well-defined theoretical connection to the formulation of graph OOD generalization outlined

in Section 3. This approach effectively captures invariant knowledge while mitigating spurious correlations, thereby facilitating robust graph OOD generalization learning. Based on the illustration above, we propose the **L**earnable **R**andom **W**alk for graph **OOD** generalization model (LRW-OOD). The detailed contributions of this paper are summarized below:

**Our contributions.** (1) *New Perspective.* To the best of our knowledge, this study is the first to systematically examine the impact of learnable random walk sequences on the graph OOD generalization problems. Our findings offer valuable insights into graph OOD learning, contributing to a deeper understanding of how random walk sequences can enhance model's performance on datasets under diverse distribution shifts. (2) *New Graph OOD Learning Paradigm.* We propose LRW-OOD, which employs an OOD-aware LRW encoder to adaptively sample sequences that adhere to specific graph OOD principles. This approach enables the model to extract random walk paths that encapsulate sufficient invariant knowledge while effectively eliminating spurious correlations. This novel paradigm offers significant insights into the advancement of graph OOD learning, paving the way for future research in this domain. (3) *SOTA Performance.* We conduct a series of performance evaluations on seven benchmark datasets, comparing our proposed model, LRW-OOD, against nine state-of-the-art graph OOD generalization models. The experimental results demonstrate that the proposed LRW-OOD outperforms the most competitive baselines when utilizing both GCN and GAT as the GNN backbone, achieving an average improvement of 3.87% on graph OOD generalization.

## 2 PRELIMINARIES

### 2.1 PROBLEM FORMULATION

**The Semi-supervised Node Classification Tasks on Graphs.** We consider a general graph representation method, denoted as $\mathbf{G} = (\mathcal{V}, \mathcal{E})$, where $|\mathcal{V}| = n$ represents the number of nodes and $|\mathcal{E}| = m$ denotes the number of edges. The adjacency matrix (including self-loops) for the graph is $\mathbf{A} \in \mathbb{R}^{n \times n}$, where each entry $\mathbf{A}(u, v) = 1$ if $(u, v) \in \mathcal{E}$ and $\mathbf{A}(u, v) = 0$ otherwise. Additionally, the node feature matrix is represented as $\mathbf{X} = \{x_1, \ldots, x_n\}$, where each $x_v \in \mathbb{R}^f$ corresponds to the feature vector associated with node $v$. The node label is denoted as $\mathbf{y}$. In the semi-supervised node classification paradigm, the graph is partitioned into a labeled node set $\mathcal{V}_L$ and an unlabeled set $\mathcal{V}_U$. The classification process leverages the graph topology and the node features, wherein the model is trained using $\mathcal{V}_L$ and subsequently applied to infer the labels of nodes in $\mathcal{V}_U$.

**The OOD Generalizations on Graphs.** In this paper, we primarily investigate the problem of graph OOD generalization at the node level, where multiple distribution shifts—such as topology shifts and node feature shifts—occur between the training and testing sets. Formally, this is characterized by a discrepancy in the joint distribution, i.e., $\mathrm{Pr}_{tr}(\mathbf{G}, \mathbf{y}) \neq \mathrm{Pr}_{te}(\mathbf{G}, \mathbf{y})$, where the objective is to accurately predict node labels despite these distribution shifts. The key factors driving such distribution shifts are referred to as confounders or environments, which can be understood through the lenses of data generation distributions and causal inference learning. Since the training and testing sets are derived from distinct environments, spurious confounders may be embedded within the correlations between the graph $\mathbf{G}$ and the label $\mathbf{y}$. However, despite the presence of these confounding factors, certain invariant or stable properties persist across different environments. The goal of graph OOD generalization models is to learn representations that capture these invariant factors while eliminating the influence of spurious confounders, thereby enhancing model performance under diverse distribution shifts.

### 2.2 RELATED WORKS

**Graph Neural Networks.** GNNs have garnered significant attention due to their efficacy in learning high-quality representations from graph-structured data (Kipf and Welling (2017); Veličković et al. (2018); Zhang et al. (2019; 2022); Klicpera et al. (2019); Xu et al. (2019)). While extensive research has been conducted on the expressiveness and representational power of GNNs, their generalization capability remains an open question, particularly in scenarios where the test data is drawn from distributions different from those of the training data (Arjovsky et al. (2019); Koyama and Yamaguchi (2020); Chen et al. (2022b)). Following the ideas of invariant risk minimization and structural causal models which reveal that the fundamental challenge of OOD generalization in graph data stems from latent confounder, this paper propose a theoretically grounded model designed to effectively

extract the invariant knowledge and discard the spurious correlations across datasets under various distribution.

**OOD Generalization Learning on Graphs.** The problem of learning under distribution shifts in graph-structured data has increasingly attracted attention within the graph learning research community. SRGNN (Zhu et al. (2021)) attempts to address performance degradation by incorporating a regularization term and reducing the disparity between embeddings derived from the training and testing sets. DGNN (Guo et al. (2024)) conducts extensive empirical studies and leverages self-attention mechanisms along with a decoupled architecture to facilitate OOD generalization in graph learning. However, these approaches fail to account for the essential factors contributing to the performance deterioration of GNNs under diverse distribution shifts. Consequently, their improvements remain limited compared to traditional GNNs. Recent advancements in addressing graph OOD generalization learning have predominantly centered on the core principles of IRM and SCMs, and can be broadly classified into two primary approaches: invariant topolog extraction and invariant spectrum extraction. Specifically, methods such as EERM (Wu et al. (2022)), CIT (Xia et al. (2023)), MARIO (Zhu et al. (2024)), CaNet (Wu et al. (2024)), GRM (Wang et al. (2025)) and DNRL (Qiao et al. (2025)) adopt the first approach by leveraging various mechanisms, including pseudo-environment-generatio, node clustering and contrastive learning, to facilitate the extraction of invariant local topological structures of nodes. Alternatively, SpCo (Liu et al. (2022)) follows the second approach by distinguishing between low-frequency and high-frequency components of the graph spectrum, treating the former as invariant knowledge and the latter as spurious information. This distinction enables the model to enhance OOD generalization through graph contrastive learning. However, as discussed in Section 1, the existence of a universally invariant topology or spectrum is not guaranteed across all graphs in OOD generalization problems. These limitations hinder the performance and generalization capability of existing models, particularly in real-world applications.

## 3 METHODS

### 3.1 MOTIVATION

As discussed in Section 1, existing graph OOD generalization models exhibit the following limitations: (1) the assumption of an invariant graph topology is not necessarily valid across graphs under different distributions; and (2) the extraction of a universally invariant graph spectrum remains unreliable. These limitations give rise to the following question: what constitutes an appropriate instantiation of invariant knowledge across graphs under multiple distribution shifts?

While existing approaches predominantly rely on either topology or spectrum as the carrier of invariant knowledge, we argue that learnable random walk sequences, sampled according to specific probabilistic rules, provide a more suitable representation of invariant knowledge in graph OOD scenarios. The motivations behind this claim is outlined as follows:

**Motivation 1:** In contrast to existing approaches where neither topology nor spectrum necessarily remains invariant under various distribution shifts, we posit that invariant knowledge can be instantiated in the form of learnable random walk sequences. This perspective is grounded in the observation that invariant knowledge can be encoded within the probability of transitioning to the next node that shares similar semantic information and possesses invariant features. As long as graphs exhibit common invariant patterns across distribution shifts, this property ensures that random walk sequences can effectively capture and preserve the underlying invariant knowledge.

**Motivation 2:** Furthermore, random walk sequences maintain a well-defined theoretical relationship with the formulation of graph OOD generalization. Specifically, the probability transition matrix governing these sequences can be learned in accordance with the principles underlying graph OOD generalization, ensuring the adaptability and robustness across various distribution shifts of graphs.

Building upon the prior works (Wu et al. (2022); Xia et al. (2023); Liu et al. (2022); Wu et al. (2024); Zhu et al. (2024)), the problem of OOD generalization in graphs can be reformulated as an optimization task. Specifically, it involves minimizing the loss of the worst-case performance of the model across multiple graphs within all possible environments. These environments comprise both invariant knowledge shared among them and spurious correlations unique to each specific

environment. This problem can be expressed as follows:

$$\min_f \max_{\mathbf{G}_e \sim \mathcal{G}} \mathcal{L}(f(\mathbf{G}_e), \mathbf{y}), \tag{1}$$

where $\mathcal{G}$ is the set of graphs under all environments, $\mathbf{G}_e$ is the graph under the environment $e$, $\mathcal{L}$ is the loss function, $f$ is the graph OOD model and $\mathbf{y}$ is the label. However, this optimization formulation cannot be directly applied to graph OOD generalization due to the inaccessibility of environmental factors $e$. To address this challenge, Wu et al. (2022) introduces two conditions that are theoretically equivalent to the aforementioned graph OOD generalization formulation while also being directly applicable to graph OOD learning. These conditions are formally stated as follows: (1) sufficient condition: $\mathbf{y} = f^*(\mathbf{G}_e) + \sigma \leftrightarrow \max_f I(\mathbf{y}, f)$, where $f^*$ denotes the model $f$ with the optimal parameters, while $\sigma$ represents the random variable entirely independent of the predicted label $\mathbf{y}$. (2) invariance condition: $\Pr(\mathbf{y}|e) = \Pr(\mathbf{y}) \leftrightarrow \min_f I(\mathbf{y}, e|f)$.

It is important to note that random walk sequences can inherently incorporate the two graph OOD generalization conditions by parameterizing the probability transition matrix in accordance with these principles. Specifically, rather than adopting the conventional approach of fixing the probability transition matrix as the degree-normalized adjacency matrix (Xie et al. (2023); Su et al. (2024)), it can instead be parameterized as a learnable matrix. This matrix is initialized using the cosine similarity matrix, which captures the semantic similarity between the soft embeddings of node pairs within graphs, thereby integrating node semantic information into the random walk sequences. However, the semantic similarity matrix alone cannot reliably capture the invariant knowledge across graphs under multiple distribution shifts, as it is susceptible to spurious correlations arising in different environments. To address this limitation, the learnable probability transition matrix must be guided by a distribution-invariant loss that adheres to the two graph OOD generalization conditions outlined above. The details of this approach will be elaborated in the following section.

## 3.2 MODEL FRAMEWORK

The overall framework of the proposed LRW-OOD method is illustrated in Figure 2 and pseudo code Algorithm 1. Our approach adopts a two-stage training paradigm for OOD generalization in graph learning: (1) the initial stage focuses on training an OOD-aware Learnable Random Walk (LRW) encoder, and (2) the subsequent stage involves training a GNN-based classifier. The LRW encoder is designed to produce high-quality node embeddings that capture invariant features across diverse distribution shifts. After training the LRW encoder, the generated embeddings are then utilized by the GNN classifier—built upon standard GNN backbones such as GCN (Kipf and Welling (2017)) or GAT (Veličković et al. (2018))—and trained by standard cross-entropy loss—through a weight-free embedding aggregator (i.e., mean-pooling or concatenation) to support various downstream tasks. Given that the primary objective of this paper is to enable graph OOD generalization learning via learnable random walk sequences, the subsequent of the section will primarily focus on the detailed formulation and design of the proposed OOD-aware LRW encoder.

**OOD-aware Learnable Random Walk Encoder.** To enable the extraction of invariant knowledge from graphs under various distribution shifts, we introduce the OOD-aware Learnable Random Walk (LRW) encoder, depicted in the top-left corner of Figure 2. Departing from conventional random walk strategies—which typically utilize a fixed degree-normalized adjacency matrix as the transition probability matrix (Xie et al. (2023); Su et al. (2024))—we propose a more adaptive approach. Specifically, the LRW sampler employs a GNN (i.e. a two-layer MLP for simplicity in the experiments) to generate soft node embeddings from the original node features. These embeddings are then used to construct a similarity-based transition matrix, which guides the sampling of $k$ random walk paths. This mechanism enables the integration of both topological and semantic information inherent in the graph data. Formally, for a given node $v_i$, and given the graph topology $\mathbf{A}$ and node features $\mathbf{X}$, the LRW sampler generates $k$ random walk paths $\{p_i^r\}_{r=1}^k$ as follows:

$$\{p_i^r\}_{r=1}^k = \text{RW}(v_i, k), \quad \text{s.t.} \quad \Pr_{v_j \in \mathcal{N}(v_i)}(v_i \to v_j) = \cos(\mathbf{z}_i, \mathbf{z}_j), \quad \mathbf{z} = \text{GNN}(\mathbf{A}, \mathbf{X}), \tag{2}$$

where $\mathcal{N}(v_i)$ denotes the set of neighboring nodes of $v_i$, and $\text{RW}(v_i, k)$ represents the random walk sampler that generates $k$ random walk paths initiated from node $v_i$. After obtaining $k$ random walk paths started at node $v_i$, we then apply a path encoder (i.e., MLP) to transform the embeddings of nodes along the $r$-th random walk path into the LRW embeddings $\mathbf{h}_i^r$:

$$\mathbf{h}_i^r = \text{MLP}(\text{Concat}(\{\mathbf{z}_j\}_{v_j \in p_i^r})), \quad \text{s.t.} \quad 1 \leq r \leq k. \tag{3}$$

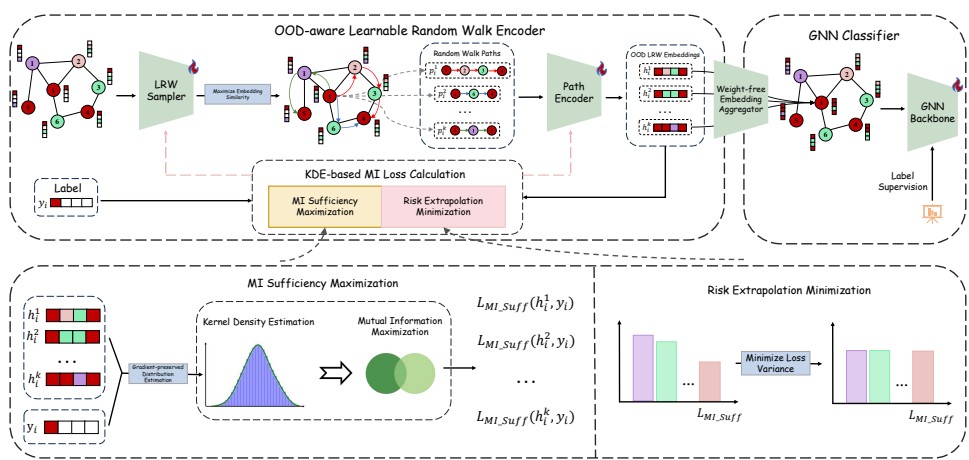

Figure 2: The framework of the proposed LRW-OOD.

The resulting LRW embeddings $\{\mathbf{h}_i^r\}_{r=1}^k$, together with the corresponding label $\mathbf{y}_i$ of node $v_i$, are subsequently input into a KDE-based MI loss module. This component serves as a key component within the LRW encoder, facilitating the model's ability to capture invariant knowledge and discard spurious correlations under OOD scenarios.

**KDE-based MI Loss Calculator.** As discussed in Section 3.1, the graph OOD generalization can be formulated as an optimization task aimed at minimizing the loss under the worst-case distribution shift. Also, this objective is theoretically equivalent to the two OOD conditions: (1) sufficiency condition, expressed as $\mathbf{y} = f^*(\mathbf{G}_e) + \sigma \leftrightarrow \max_f I(\mathbf{y}, f)$, and (2) invariance condition, given by $\Pr(\mathbf{y}|e) = \Pr(\mathbf{y}) \leftrightarrow \min_f I(\mathbf{y}, e|f)$. To encourage the LRW encoder to generate node embeddings that capture invariant features while discarding spurious correlations, we introduce two complementary objectives: the *MI sufficiency maximization loss* and the *risk extrapolation minimization loss*, which correspond to the sufficiency and invariance conditions, respectively.

To achieve the sufficiency condition, a natural approach would be to directly maximize the mutual information (Steuer et al. (2002); Kraskov et al. (2004)) between the LRW embeddings $\{\mathbf{h}_i^r\}_{r=1}^k$ and the corresponding label $\mathbf{y}_i$. However, this approach is unfeasible due to the agnosticism of the latent label distribution $\Pr(\mathbf{y})$ in the context of graph OOD generalization (Wu et al. (2024); Zhu et al. (2024)). As a result, traditional graph OOD models often resort to distance-based loss functions, such as KL divergence, to approximately satisfy the sufficiency condition (Wu et al. (2022); Wang et al. (2025)). Nevertheless, these methods are prone to inaccuracies due to the approximations they involve. In contrast, our proposed MI sufficiency maximization approach leverages an estimation-based mechanism to approximate the distributions of the LRW embeddings $\{\Pr(\mathbf{h}_i^r)\}_{r=1}^k$ and the label distribution $\Pr(\mathbf{y})$, allowing the model to directly compute and maximize the mutual information between these variables. Rather than relying on histogram-based estimation methods—which are non-differentiable due to the discrete partitioning of the hypercubes—we utilize kernel density estimation (KDE) with a Gaussian kernel (Moon et al. (1995); Steuer et al. (2002); Kraskov et al. (2004)). This choice is motivated by the differentiability of KDE and its computational efficiency, which allows for gradient-based optimization. The formal expression of this approach is provided below:

$$\mathcal{L}_{\mathbf{MI\_Suff}}(\mathbf{h}_i^r, \mathbf{y}_i) = -\hat{\Pr}(\mathbf{h}_i^r, \mathbf{y}_i) \log_2 \frac{\hat{\Pr}(\mathbf{h}_i^r, \mathbf{y}_i)}{\hat{\Pr}(\mathbf{h}_i^r)\hat{\Pr}(\mathbf{y}_i))},$$

$$\text{s.t.} \quad \hat{\Pr}(\mathbf{s}) = \frac{1}{k}\sum_{r=1}^k K(\mathbf{u}) = \frac{1}{k}\sum_{r=1}^k \frac{e^{-\mathbf{u}/2}}{(2\pi)^{d/2}m^d det(\mathbf{S}^{1/2})},$$

(4)

where $d$ represents the dimension of the random variable $\mathbf{s}$, $m$ denotes the kernel bandwidth, $\mathbf{S} = m^2 I_d$ is the kernel covariance matrix associated with the kernel bandwidth $m$, $\mathbf{u} = \frac{(\mathbf{s}-\mathbf{s}^r)^T \mathbf{S}^{-1}(\mathbf{s}-\mathbf{s}^r)}{m^2}$ represents the Mahalanobis distance (McLachlan (1999); De Maesschalck et al. (2000)) between the

variable $\mathbf{s}$ and the sample $\mathbf{s}^r$, and $K(\mathbf{u})$ is the kernel function (i.e., Gaussian kernel in our paper). Meanwhile, following the prior works (Wu et al. (2022); Wang et al. (2025)), we introduce the risk extrapolation minimization loss by minimizing the variance among the MI sufficiency maximization losses $\{\mathcal{L}_{\mathrm{MI\_Suff}}(\mathbf{h}_i^r, \mathbf{y}_i)\}_{r=1}^k$. Based on the illustration above, we propose the overall KDE-based MI loss formulated as below:

$$\mathcal{L} = \sum_{i=1}^n \left( \mathbb{V}(\{\mathcal{L}_{\mathrm{MI\_Suff}}(\mathbf{h}_i^r, \mathbf{y}_i)\}_{r=1}^k) + \frac{1}{k} \sum_{r=1}^k \mathcal{L}_{\mathrm{MI\_Suff}}(\mathbf{h}_i^r, \mathbf{y}_i) \right). \tag{5}$$

### 3.3 Theoretical Discussion

In this section, we will illustrate the theoretical guarantee that the proposed KDE-based MI loss formulated as Eq. (5) is capable of guiding a valid random walk sampling for graph OOD generalization formulated as Eq. (1). Moreover, we will also give a theoretical upper bound of LRW-OOD, showcasing the computation-friendly property of the proposed model. The detailed proofs for the following theorems are illustrated in the Appendix C.

**Theorem 3.1.** *Let $f(\mathbf{G}_e)$ denotes the learnable random walk encoder. If it is optimized by minimizing the KDE-based MI loss defined in Eq. (5), then the resulting encoder satisfies both the sufficiency condition: $\mathbf{y} = f^*(\mathbf{G}_e) + \sigma$ and the invariance condition: $\Pr(\mathbf{y}|e) = \Pr(\mathbf{y})$.*

Building upon Theorem 3.1, we establish the theoretical guarantee that connects the OOD conditions to the formulation of graph OOD generalization through the following theorem:

**Theorem 3.2.** *Let $f^*(\mathbf{G}_e)$ denotes the optimized learnable random walk encoder satisfying both the sufficiency and invariance conditions. Then, the encoder $f^*(\mathbf{G}_e)$ is the solution to the graph OOD generalization formulated as Eq. (1).*

Building upon Theorem 3.1 and Theorem 3.2, we further discuss the theoretical upper bound of the OOD error of the proposed KDE-based MI loss through the following theorem:

**Theorem 3.3.** *Let $n, k$ be the number of the nodes and the sampled random walk paths, $\mathbf{h}_i^r$ be the $r$-th LRW embedding of node $v_i$ and $d$ be the dimension of the LRW embedding $\mathbf{h}_i^r$. Then, the proposed KDE-based MI loss is convergent.*

The preceding theorems establish that the learnable random walk encoder, optimized by the KDE-based MI loss, is theoretically capable of generating random walk sequences that preserve invariant information while effectively eliminating spurious correlations across datasets exhibiting distributional shifts. Furthermore, with the OOD-awareness guaranteed by theorems above, we proceed to demonstrate the model's time efficiency and memory efficiency through the following theorem:

**Theorem 3.4.** *Let $n, k, s, d$ be the number of nodes, the times of random walk per node, the walk length per node and the feature dimension, $l_1$ be the number of layers for the LRW sampler, and $l_2$ be the number of layers for the path encoder. The overall time complexity and space complexity of LRW-OOD is $\mathcal{O}\left(nd^2(l_1 + l_2) + nksd\right)$ and $\mathcal{O}\left(nd(l_1 + l_2)\right)$, respectively.*

## 4 Experiments

In this section, we present comprehensive experiments evaluating the proposed LRW-OOD under datasets with diverse distribution shifts. The objective of these experiments is to address the following research questions: **Q1**: How does the proposed model perform compared to state-of-the-art models on datasets under various distribution shifts? **Q2**: To what extent do the proposed components of LRW-OOD contribute to its OOD generalization capabilities on graph data? **Q3**: How sensitive is the performance of the proposed model to variations in its hyperparameters? **Q4**: What insights can be obtained from visualizing the representations of the proposed model? Due to page limit, we put part of experiment results and detailed analysis in Appendix E.

### 4.1 Experiment Setup

**Datasets.** In line with prior studies, we employ seven node classification datasets that exhibit diverse sizes, characteristics, and types of distribution shifts. These datasets are categorized as follows:

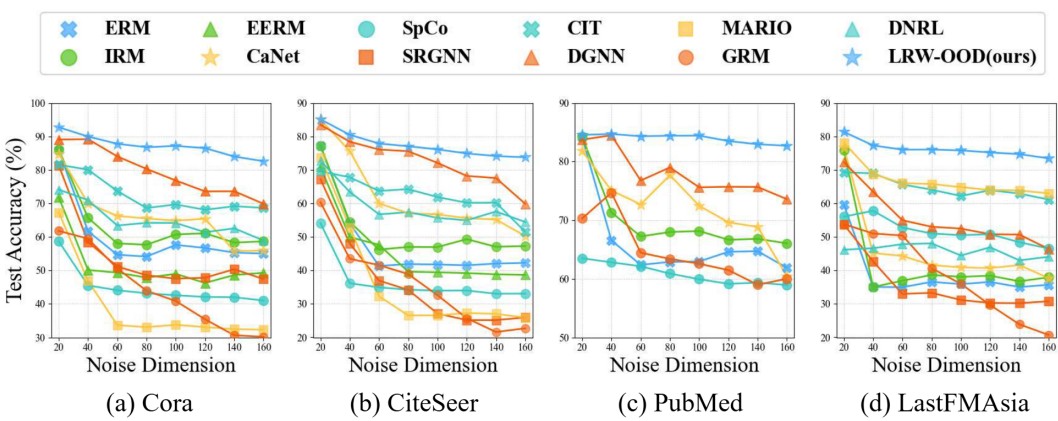

Figure 3: The performance comparison of graph OOD models using GCN as the backbone.

(1) synthetic datasets: Cora (Yang et al. (2016)), CiteSeer (Yang et al. (2016)), PubMed (Yang et al. (2016)), and LastFMAsia (Rozemberczki and Sarkar (2020)); (2) cross-domain datasets: Twitch (Rozemberczki et al. (2021)) and WebKB (Pei et al. (2020)); and (3) temporal evolution dataset: ogb-ArXiv (Hu et al. (2020)). Specifically, for the synthetic datasets, we augment the original node features with the artificially generated noise of environment-specific spurious correlations, whose dimension $d_{\text{spu}}$ varies from 20 to 160. For cross-domain datasets, we construct the training, validating and testing sets by dividing the whole graph into three subgraphs with distinct domain meanings. For the temporal dataset, we choose papers that were published before 2017 as training set, those published in 2018 as validation set and the remaining for the testing set. A comprehensive summary of the datasets is provided in Table 2, with additional details about data pre-preprocessing presented in Appendix D.

**Baselines.** We evaluate the performance of the proposed LRW-OOD model against 11 state-of-the-art graph OOD generalization methods, namely ERM, IRM, SRGNN, SpCo, EERM, CIT, DGNN, CaNet, MARIO, GRM and DNRL. To mitigate the effects of randomness and ensure reliable evaluation, each experiment is repeated 10 times, and the average performance is reported.

**Experiment Environment.** To facilitate reproducibility, we report the hardware and software configurations used in our experiments. All experiments were conducted on a server equipped with an Intel(R) Xeon(R) Gold 6240 CPU 2.60GHz, and a NVIDIA A800 GPU with 80GB memory, utilizing CUDA version 12.4.

### 4.2 PERFORMANCE COMPARISON

To answer **Q1**, we evaluate our model against baselines on datasets with synthetic, cross-domain, and temporal shifts, with results in Figures 3, 4, and Table 1.

Table 1: The performance of graph OOD models on corss-domain and temporal evolution datasets.

| Dataset | Backbone | ERM | IRM | EERM | SpCo | SRGNN | CIT | CaNet | DGNN | MARIO | GRM | DNRL | LRW-OOD |
|---------|----------|-----|-----|------|------|-------|-----|-------|------|-------|-----|------|---------|
| Twitch | GCN | 52.1±1.6 | 52.1±1.7 | OOM | 49.0±0.4 | 47.8±2.4 | 53.9±0.3 | 54.3±0.8 | 52.7±3.1 | 53.9±0.1 | 55.1±2.7 | 46.2±0.1 | **55.5±1.0** |
|  | GAT | 51.8±1.0 | 51.5±1.0 | OOM | 46.9±1.6 | 53.9±0.1 | 54.0±0.4 | 54.0±0.2 | 53.8±0.2 | 54.4±0.3 | 53.9±0.1 | 46.1±0.1 | **55.3±1.3** |
| WebKB | GCN | 9.8±0.1 | 9.8±0.1 | 28.4±10.3 | 9.8±0.1 | 9.9±0.2 | 16.0±1.0 | 11.9±5.7 | 48.8±12.7 | 40.4±18.2 | 41.6±7.5 | 9.8±0.1 | **52.0±1.9** |
|  | GAT | 9.9±0.2 | 10.6±2.0 | 29.5±18.2 | 15.3±2.9 | 56.5±0.9 | 9.8±0.1 | 19.5±9.4 | 37.7±21.9 | 41.4±17.3 | 48.7±6.3 | 9.8±0.1 | **56.7±3.1** |
| ogb-ArXiv | GCN | 52.7±0.2 | 53.1±0.4 | 39.7±1.4 | OOM | 48.3±0.5 | OOM | 52.8±2.0 | 44.8±0.6 | OOM | 49.0±4.6 | OOM | **57.7±0.1** |
|  | GAT | 52.8±0.2 | 53.9±0.3 | 46.7±0.5 | OOM | 49.2±0.7 | OOM | 57.1±2.7 | 43.1±2.1 | OOM | 54.3±1.0 | OOM | **58.9±0.1** |

**Distribution Shifts on Synthetic Datasets.** We evaluate testing accuracy on Cora, CiteSeer, PubMed, and LastFMAsia, each augmented with spurious noise features of dimensions 20-160 (Figures 3 and 4). Although all models degrade with increasing noise dimensionality, LRW-OOD consistently

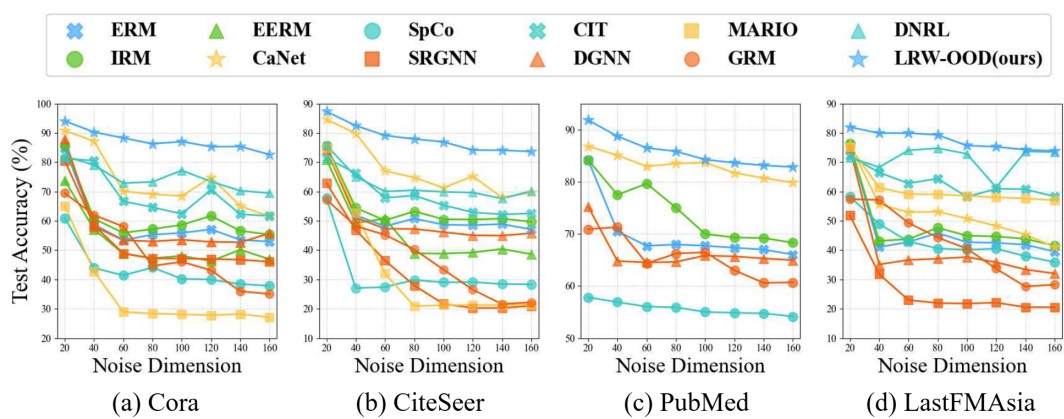

Figure 4: The performance comparison of graph OOD models using GAT as the backbone.

outperforms baselines with both GCN and GAT backbones, demonstrating strong robustness to distribution shifts.

**Distribution Shifts on Cross-domain & Temporal Evolution Datasets.** As shown in Table 1, LRW-OOD consistently outperforms all baselines on WebKB, Twitch, and ogb-ArXiv, despite substantial domain and temporal shifts. It achieves average gains of 3.1% with GCN and 1.0% with GAT, demonstrating its capacity to learn invariant representations and mitigate spurious correlations, thereby supporting robust real-world deployment.

### 4.3 ABLATION STUDY

To answer **Q2**, we conduct an ablation study of LRW-OOD on Cora, CiteSeer, PubMed, and LastFMAsia, with results reported in Table 3 (Appendix E). The variants include: w/o SM, replacing the KDE-based sufficiency maximization loss with KL-divergence (Wu et al. (2022); Wang et al. (2025)); w/o REM, removing the risk extrapolation minimization loss; w/o KDE_MI, removing the KDE-based MI loss and w/o LRW, the baseline random-walk GNN without the LRW encoder. Results show that w/o LRW yields the relatively poorest performance, underscoring the encoder's critical role in extracting invariant knowledge. Moreover, w/o REM generally outperforms w/o SM, highlighting the greater importance of SM loss for graph OOD generalization.

### 4.4 SENSITIVITY ANALYSIS

To answer **Q3**, we examine the effect of two hyperparameters—the number of walk steps and walk times—on LRW-OOD, with results in Tables 4 and 5 (Appendix E). Table 4 shows that a single step suffices for Cora, CiteSeer, and PubMed, while LastFMAsia requires longer walks due to lower homophily, where invariant patterns extend beyond 1-hop neighborhoods. Table 5 further indicates that multiple walks per node consistently improve performance, as individual walks may capture spurious correlations, whereas multiple walks preserve invariant patterns and mitigate noise.

### 4.5 MODEL VISUALIZATION

To address **Q4**, we visualize LRW embeddings from different walk orders on Cora, CiteSeer, PubMed, and LastFMAsia (Figure 5 in Appendix E). The embeddings show clear distinctions across datasets with varying spurious correlations. This indicates that LRW encodes distinct representations from random walks, mitigating spurious correlations and capturing invariant information, thereby yielding more expressive representations for graph OOD generalization.

## 5  CONCLUSION

In this paper, we introduce LRW-OOD, a novel approach for graph OOD generalization at the node level, which necessitates the model's ability to handle multiple distribution shifts between the training set and the testing set. Distinct from existing methods that primarily rely on graph topology or spectral properties as the medium for invariant knowledge, our method leverages Learnable Random Walk (LRW) sequences to capture such invariant representations. Rather than utilizing a conventional fixed-probability transition matrix (e.g., the degree-normalized adjacency matrix), our framework employs an LRW-based sampler alongside a path encoder to learn LRW embeddings that parameterize the transition probabilities of the random walk. To ensure the generated random walk sequences conform to the OOD principles, we further propose a KDE-based MI loss, which integrates an MI sufficiency maximization component and a risk extrapolation minimization component. Extensive experimental evaluations demonstrate the superior performance of LRW-OOD in addressing diverse types of distribution shifts across various graph datasets.

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

---

**Algorithm 1** Our Proposal's LRW-OOD Workflow

---

1: **for** training epoch $l = 1, \cdots, L$ **do**
2:     Generate node embeddings $\mathbf{z}$ through a GNN encoder according to Eq. (2);
3:     **for** each node $i = 1, \cdots, n$ **do**
4:         Calculate the transition probability from node $v_i$ to its neighbor $v_j$ according to Eq. (2);
5:         Sample $k$ random walk paths starting at node $v_i$ according to Eq. (2);
6:         **for** each path $r = 1, \cdots, k$ **do**
7:             Obtain the LRW embedding $\mathbf{h}_i^r$ of $r$-th path starting at node $v_i$ according to Eq. (3);
8:             Calculate the MI sufficient loss of between LRW path embedding $\mathbf{h}_i^r$ and node label $\mathbf{y}_i$ according to Eq. (4);
9:         **end for**
10:    **end for**
11:    Calculate the overall KDE-based MI loss according to Eq. (5);
12:    Update the parameters in the LRW sampler and path encoder according to the gradient calculated from Eq. (5);
13: **end for**

---

## A   The Overall Workflow of the Proposed LRW-OOD

The overall workflow of the proposed LRW-OOD framework is depicted in Figure 2, with the corresponding pseudocode provided in Algorithm 1. During each of the $L$ training epochs, node embeddings are first generated using a GNN encoder, as defined in Eq. (2). Subsequently, for each node $v_i$ ($i = 1, \ldots, n$) in the graph, the transition probabilities to its neighboring nodes $v_j$ are computed. Based on these probabilities, $k$ random walk paths originating from node $v_i$ are sampled in accordance with Eq. (2). Finally, the model computes a KDE-based MI loss and updates the parameters of both the LRW sampler and the path encoder using the gradient of the loss, as specified in Eq. (5).

## B   Case Study: Detailed Explanation about the Proposed Method Solving the Dilemma from Figure 1

As illustrated in **L1** and **L2** of Section 1, both topology-based models and spectrum-based models fail to solve the dilemma of the case depicted in Figure 1, due to the facts that: (1) if subgraphs are used as carriers of invariant knowledge, the model may fail to accurately extract such invariant subgraphs with similar topological structures but different semantics; (2) if graph spectra are used as carriers of invariant knowledge, the model may fail to accurately extract invariant spectra among graphs with strong homophily and those with strong heterophily due to the underlying temporal distribution shift.

Different from these models above, the proposed model based on learnable random walk sequences is capable of integrating graph topology and node features together. Thus, it can successfully extract the invariant knowledge within the heterophilic citation networks under temporal distribution shift in Figure 1. To illustrate, taking Figure 1 as an example, during the early stages of training, the LRW sampler performs random walks based on initial feature similarity, which tends to aggregate information from homophilic neighbors. In the case of the 1990s citation network, physical nodes tend to aggregate information from other physical nodes, often ignoring mathematical nodes. However, as the training process continues, the KDE-based MI loss function, based on OOD principles, helps the LRW sampler identify spurious correlations in the initial probability transition matrix, which was based on feature similarity. As a result, the LRW sampler dynamically adjusts the features of each node to retain the invariant information and discard the spurious correlations. This allows the model to aggregate information from math nodes in the 2020s citation network, enabling correct node classification tasks in the 2020s networks.

## C   Theoretical Analysis for Section 3.3

Before proceeding with the proofs of the theorems presented in Section 3.3, we first introduce a useful lemma concerning the theoretical effectiveness of kernel density estimation:

**Lemma C.1.** *Let $I(\mathbf{x}, \mathbf{y})$ be the real mutual information of the random variables $\mathbf{x}, \mathbf{y}$, and $\hat{I}(\mathbf{x}, \mathbf{y})$ be the corresponding kernel density estimation as defined in Eq. 4, then $\hat{I}(\mathbf{x}, \mathbf{y})$ converges in probability to $I(\mathbf{x}, \mathbf{y})$, i.e.:*

$$\hat{I}(\mathbf{x}, \mathbf{y}) \xrightarrow{a.s.} I(\mathbf{x}, \mathbf{y}). \tag{6}$$

*Proof.* Recall from Section 3.3 that the kernel density function we employ is Gaussian kernel function formulated as: $\hat{\Pr}(\mathbf{x}) = \frac{e^{-u^2/2}}{(2\pi)^{d/2} m^d det(\mathbf{X}^{1/2})}$. According to the propositions of Gaussian kernel function (Silverman (2018); Moon et al. (1995)), we have:

$$\int_{-\infty}^{+\infty} \frac{e^{-u^2/2}}{(2\pi)^{d/2} m^d det(\mathbf{X}^{1/2})} du = 1,$$

$$\int_{-\infty}^{+\infty} u \frac{e^{-u^2/2}}{(2\pi)^{d/2} m^d det(\mathbf{X}^{1/2})} du = 0, \tag{7}$$

$$\int_{-\infty}^{+\infty} u^2 \cdot \frac{e^{-u^2/2}}{(2\pi)^{d/2} m^d det(\mathbf{X}^{1/2})} du = 1 < +\infty,$$

which signify the following facts: (1) the integration of Gaussian kernel function is 1; (2) the bias of Gaussian kernel function is 0; and (3) The variance of Gaussian kernel function is bounded. Thus, according to the theorem of Glivenko–Cantelli (Ash and Doléans-Dade (2000)), we have:

$$\sup_{\mathbf{x}, \mathbf{y}} |\hat{\Pr}(\mathbf{x}, \mathbf{y}) - \Pr(\mathbf{x}, \mathbf{y})| \xrightarrow{a.s.} 0,$$

$$\sup_{\mathbf{x}} |\hat{\Pr}(\mathbf{x}) - \Pr(\mathbf{x})| \xrightarrow{a.s.} 0, \tag{8}$$

$$\sup_{\mathbf{y}} |\hat{\Pr}(\mathbf{y}) - \Pr(\mathbf{y})| \xrightarrow{a.s.} 0,$$

which illustrate that the distributions and the joint distribution of random variable $\mathbf{x}, \mathbf{y}$ all converge in probability. Given that the continuous form of mutual information $I(\mathbf{x}, \mathbf{y}) = \int_{\mathbf{x}} \int_{\mathbf{y}} \Pr(\mathbf{x}, \mathbf{y}) \log_2 \frac{\Pr(\mathbf{x}, \mathbf{y})}{\Pr(\mathbf{x})\Pr(\mathbf{y})} d\mathbf{x} d\mathbf{y}$ and $\hat{I}(\mathbf{x}, \mathbf{y}) = \int_{\mathbf{x}} \int_{\mathbf{y}} \hat{\Pr}(\mathbf{x}, \mathbf{y}) \log_2 \frac{\hat{\Pr}(\mathbf{x}, \mathbf{y})}{\hat{\Pr}(\mathbf{x})\hat{\Pr}(\mathbf{y})} d\mathbf{x} d\mathbf{y}$, the formulations of these two mutual information also converge in probability. Thus, the lemma is proved. $\square$

### C.1 THEORETICAL PROOFS FOR THEOREM 3.1

For the sufficiency condition, we have the following fact that minimizing $\mathcal{L}_{\mathbf{MI\_Suff}}(\mathbf{h}, \mathbf{y}) = -\hat{\Pr}(\mathbf{h}, \mathbf{y}) \log_2 \frac{\hat{\Pr}(\mathbf{h}, \mathbf{y})}{\hat{\Pr}(\mathbf{h})\hat{\Pr}(\mathbf{y}))}$ is equivalent to maximizing the discrete form of mutual information $I(\mathbf{h}, \mathbf{y})$, according to lemma C.1. Let $\mathbf{h}^*$ be the optimal parameters that satisfy the sufficiency condition $\mathbf{y} = f^*(\mathbf{G}_e) + \sigma$, and $\mathbf{h}'$ be the parameters that are learned by minimizing $\mathcal{L}_{\mathbf{MI\_Suff}}(\mathbf{h}, \mathbf{y})$. Suppose $\mathbf{h}^* \neq \mathbf{h}'$. Then, according to the definitions of $\mathbf{h}^*, \mathbf{h}'$ and the propositions of mutual information, we have:

$$I(\mathbf{h}^*, \mathbf{y}) = I(\mathbf{h}^*, f^*(\mathbf{h}^*) + \sigma)$$
$$= I(\mathbf{h}^*, f^*(\mathbf{h}^*)), \tag{9}$$

$$I(\mathbf{h}', \mathbf{y}) = I(\mathbf{h}', f^*(\mathbf{h}^*) + \sigma)$$
$$= I(\mathbf{h}', f^*(\mathbf{h}^*)). \tag{10}$$

Since $\mathbf{h}^* \neq \mathbf{h}'$, it's obvious that $I(\mathbf{h}^*, f^*(\mathbf{h}^*)) \geq I(\mathbf{h}', f^*(\mathbf{h}^*))$. However, $\mathbf{h}'$ is obtained by minimizing $\mathcal{L}_{\mathbf{MI\_Suff}}(\mathbf{h}, \mathbf{y})$, which suggests that $I(\mathbf{h}^*, f^*(\mathbf{h}^*)) \leq I(\mathbf{h}', f^*(\mathbf{h}^*))$. The contradiction exists due to the false prerequisite of $\mathbf{h}^* \neq \mathbf{h}'$. Thus, $\mathbf{h}^* = \mathbf{h}'$, and the sufficiency condition is satisfied.

According to the invariance condition, we have:

$$\Pr(\mathbf{y}|e) = \Pr(\mathbf{y}) \leftrightarrow \frac{\Pr(\mathbf{y}|e)}{\Pr(\mathbf{y})} = 1$$

$$\leftrightarrow \frac{\Pr(\mathbf{y}, e)}{\Pr(\mathbf{y})\Pr(e)} = 1. \tag{11}$$

Note that $I(\mathbf{y}, e) = \sum_{\mathbf{y},e} \Pr(\mathbf{y}, e) \log_2 \frac{\Pr(\mathbf{y},e)}{\Pr(\mathbf{y})\Pr(e)}$. Thus, given the fact that $I(\mathbf{y}, e) \geq 0$, Eq. 11 is equivalent to minimizing $I(\mathbf{y}, e)$. According to the chain low of mutual information, $I(\mathbf{y}, e) \leq I(\mathbf{y}, e | \mathbf{h})$. As a consequence, minimizing $I(\mathbf{y}, e | \mathbf{h})$ is equivalent to minimizing $I(\mathbf{y}, e)$. Also, given the definitions of mutual information and KL divergence, we have:

$$
\begin{aligned}
I(\mathbf{y}, e | \mathbf{h}) &= \mathrm{KL}(\Pr(\mathbf{y} | \mathbf{h}, e) || \Pr(\mathbf{y} | \mathbf{h})) \\
&= \mathrm{KL}(\Pr(\mathbf{y} | \mathbf{h}, e) || \mathbb{E}_e(\mathbf{y} | \mathbf{h}, e)) \\
&\leq \mathrm{KL}(\Pr(\mathbf{y} | \mathbf{h}) || \mathbb{E}_e(\mathbf{y} | \mathbf{h})) \\
&\leq \mathbb{V}_e(\mathbf{y} | \mathbf{h}),
\end{aligned}
\tag{12}
$$

where the first inequality is achieved due to the inaccessibility of the spurious environment $e$, and the second inequality is achieved due to the convexity of KL divergence and the Jenson Inequality. As the sufficiency condition illustrated above, the predicted label $\mathbf{y}$ is derived from the embedding $\mathbf{h}$ optimized by minimizing $\mathcal{L}_{\mathbf{MI\_Suff}}$. Thus, minimizing the risk extrapolation term $\mathbb{V}(\{\mathcal{L}_{\mathrm{MI\_Suff}}(\mathbf{h}_i^r, \mathbf{y}_i)\}_{r=1}^k)$ in the Eq. 5 is capable of minimizing $\mathbb{V}_e(\mathbf{y} | \mathbf{h})$, thus realizing the invariance condition.

## C.2 THEORETICAL PROOFS FOR THEOREM 3.2

As illustrated as the Eq. 1, the OOD formulation requires models to enhance the worst-case performance among multiple graphs within all possible latent environments. Thus, to prove theorem 3.2, we need to prove the value of the KDE-based MI loss $\mathcal{L}(f^*(\mathbf{G}_e), \mathbf{y})$ calculated under the optimal parameters $f^*$ is less than $\mathcal{L}(f(\mathbf{G}_e), \mathbf{y})$ under any sub-optimal parameters $f$, formulated as:

$$
\max_{\mathbf{G}_e \sim \mathcal{G}} \mathcal{L}(f(\mathbf{G}_e), \mathbf{y}) \geq \max_{\mathbf{G}_e \sim \mathcal{G}} \mathcal{L}(f^*(\mathbf{G}_e), \mathbf{y}).
\tag{13}
$$

The formulation above is equivalent to proving the statement that given any kinds of paramenters $f$, there exists a graph under certain environment $\mathbf{G}_e' \sim \mathcal{G}$ and the corresponding loss value of $\mathcal{L}(f(\mathbf{G}_e'), \mathbf{y})$, such that the loss value of $\mathcal{L}(f^*(\mathbf{G}_e), \mathbf{y})$ under any graph $\mathbf{G}_e \sim \mathcal{G}$ is less than that of $\mathcal{L}(f(\mathbf{G}_e'), \mathbf{y})$, formulated as below:

$$
\mathcal{L}(f(\mathbf{G}_e'), \mathbf{y}) \geq \mathcal{L}(f^*(\mathbf{G}_e), \mathbf{y}).
\tag{14}
$$

Note that the optimized parameter $f^*$ has satisfied the sufficiency condition $\mathbf{y} = f^*(\mathbf{G}_e) + \sigma$, which suggests that the loss of $f^*$ is minimized compared to other kinds of parameters $f$ under the graph $\mathbf{G}_e'$. We can derive that $\mathcal{L}(f(\mathbf{G}_e'), \mathbf{y}) \geq \mathcal{L}(f^*(\mathbf{G}_e'), \mathbf{y})$. Moreover, the optimized parameter $f^*$ has also satisfied the invariance condition $\Pr(\mathbf{y} | e) = \Pr(\mathbf{y})$, which suggests that the loss of $f^*$ is minimized under any graph with arbitrary environment $\mathbf{G}_e$. As a result, We can obtain that $\mathcal{L}(f^*(\mathbf{G}_e'), \mathbf{y}) \geq \mathcal{L}(f^*(\mathbf{G}_e), \mathbf{y})$. Based on the illustration above, the theorem 3.2 is thus proved.

## C.3 THEORETICAL PROOFS FOR THEOREM 3.3

In this section, we present a comprehensive theoretical analysis of the convergence property of the proposed KDE-based MI loss. Initially, we conduct an in-depth examination of the upper and lower bounds of the OOD error associated with the proposed loss function. Subsequently, we establish the convergence of the loss function, utilizing the derived bounds, while also analyzing the convergence rate of the kernel density estimation.

As for the upper-bound of the loss, according to the definition of $\mathcal{L}_{\mathrm{MI\_Suff}}(\mathbf{h}_i^r, \mathbf{y}_i)$ formulated as Eq. (4) and the non-negative nature of the mutual information, we have the fact that $\mathcal{L}_{\mathbf{MI\_Suff}}(\mathbf{h}_i^r, \mathbf{y}_i) \leq 0$. Thus, we have:

$$
\begin{aligned}
\mathcal{L} &= \sum_{i=1}^n \left( \mathbb{V}(\{\mathcal{L}_{\mathrm{MI\_Suff}}(\mathbf{h}_i^r, \mathbf{y}_i)\}_{r=1}^k) + \frac{1}{k} \sum_{r=1}^k \mathcal{L}_{\mathrm{MI\_Suff}}(\mathbf{h}_i^r, \mathbf{y}_i) \right) \\
&\leq \sum_{i=1}^n \mathbb{V}(\{\mathcal{L}_{\mathrm{MI\_Suff}}(\mathbf{h}_i^r, \mathbf{y}_i)\}_{r=1}^k),
\end{aligned}
\tag{15}
$$

Let $\mathcal{L}_{\text{MI\_Suff}}^{\max}, \mathcal{L}_{\text{MI\_Suff}}^{\min}$ be the maximum and the minimum value of the set $\{\mathcal{L}_{\text{MI\_Suff}}(\mathbf{h}_i^r, \mathbf{y}_i)\}_{r=1}^k$, respectively. According to the Popoviciu's inequality (Niculescu (2009); Butt et al. (2015)), we have:

$$
\begin{aligned}
\mathcal{L} &\leq \sum_{i=1}^n \mathbb{V}(\{\mathcal{L}_{\text{MI\_Suff}}(\mathbf{h}_i^r, \mathbf{y}_i)\}_{r=1}^k) \\
&\leq \sum_{i=1}^n \frac{(\mathcal{L}_{\text{MI\_Suff}}^{\max} - \mathcal{L}_{\text{MI\_Suff}}^{\min})^2}{4} \\
&\leq \sum_{i=1}^n \max\left\{(\mathcal{L}_{\text{MI\_Suff}}^{\max})^2, (\mathcal{L}_{\text{MI\_Suff}}^{\min})^2\right\},
\end{aligned}
\tag{16}
$$

Note that $\mathcal{L}_{\textbf{MI\_Suff}}(\mathbf{h}, \mathbf{y}) = -\hat{\text{Pr}}(\mathbf{h}, \mathbf{y}) \log_2 \frac{\hat{\text{Pr}}(\mathbf{h}, \mathbf{y})}{\hat{\text{Pr}}(\mathbf{h})\hat{\text{Pr}}(\mathbf{y})}$, where $\hat{\text{Pr}}(*)$ is computed through kernel density estimation. Thus, it is equivalent to the following equation: $\mathcal{L}_{\textbf{MI\_Suff}}(\mathbf{h}, \mathbf{y}) = -(I(\mathbf{h}, \mathbf{y}) + \mathcal{O}(k^{\frac{1}{\alpha}}))$, where the first term is the real mutual information between the LRW embedding $\mathbf{h}$ and the label $\mathbf{y}$, and the second term is the estimated error of the kernel density estimation (Wand and Jones (1994); Silverman (2018)). Since both $\mathbf{h}$ and $\mathbf{y}$ have the finite range, $I(\mathbf{h}, \mathbf{y})$ is equivalent to a constant $C_{\max}$ that has no relationship with the OOD error. Thus, we have:

$$
\begin{aligned}
\mathcal{L} &\leq \sum_{i=1}^n \max\left\{(\mathcal{L}_{\text{MI\_Suff}}^{\max})^2, (\mathcal{L}_{\text{MI\_Suff}}^{\min})^2\right\} \\
&\leq \sum_{i=1}^n (C_{\max}^2 + \mathcal{O}(k^{\frac{2}{\alpha}})) \\
&\leq nC_{\max}^2 + \mathcal{O}(nk^{\frac{2}{\alpha}}).
\end{aligned}
\tag{17}
$$

Similarly, as for the lower-bound of the loss, we have:

$$
\begin{aligned}
\mathcal{L} &\geq \frac{1}{k} \sum_{i=1}^n \sum_{r=1}^k \mathcal{L}_{\textbf{MI\_Suff}}(\mathbf{h}_i^r, \mathbf{y}_i) \\
&\geq \sum_{i=1}^n (C_{\min} + \mathcal{O}(k^{\frac{1}{\beta}})) \\
&\geq nC_{\min} + \mathcal{O}(nk^{\frac{1}{\beta}}).
\end{aligned}
\tag{18}
$$

Since both the upper-bound and the lower-bound are convergent to finite values when $k \to \infty$, the proposed loss function has a finite range. Also, according to Ouimet and Tolosana-Delgado (2022), the convergent rate of the estimated error of the kernel density estimation is $\mathcal{O}(k^{-\frac{4}{d+4}})$. This means that the proposed loss function is convergent when $k \to \infty$.

This property illustrates the fact that the proposed KDE-based MI loss can effectively guide the model to extract invariant knowledge while discarding the spurious correlations under various distribution shifts, as long as the sufficient random walk paths are sampled. Thus, theorem 3.3 is proved.

## C.4 Theoretical Proofs for Theorem 3.4

In this section, we provide a detailed theoretical analysis to establish the computational efficiency of the proposed model, as formalized in Theorem 3.4. Let $n$ denote the number of nodes, $k$ the number of random walks initiated per node, $s$ the length of each walk, and $d$ the dimensionality of node features. Furthermore, let $l_1$ and $l_2$ represent the number of layers in the GNN-based LRW sampler and the MLP-based path encoder, respectively. The LRW sampler utilizes a GNN with $l_1$ layers to compute LRW embeddings, incurring a time complexity of $\mathcal{O}(nd^2l_1)$. Similarly, the path encoder, implemented as an $l_2$-layer MLP, introduces an additional time complexity of $\mathcal{O}(nd^2l_2)$. Regarding the learnable random walk process, each node initiates $k$ random walks of length $s$, and at each step computes cosine similarities between its own LRW embedding and those of its neighbors. This operation results in a time complexity of $\mathcal{O}(nksd)$. Consequently, the total time complexity of the

proposed LRW-OOD model can be expressed as $\mathcal{O}\left(nd^2(l_1 + l_2) + nksd\right)$. The space complexity of LRW-OOD is composed of LRW embeddings and the path embeddings generated in the LRW sampler and the path encoder. Thus, the overall space complexity of the proposed model is $\mathcal{O}(nd(l_1 + l_2))$. Thus, Theorem 3.4 is proved.

# D    DATASETS AND PRE-PROCESSING

In this section, we describe the experimental datasets used in this paper, along with the corresponding data pre-processing procedures and dataset splitting strategies. The datasets are categorized into three distinct groups based on the type of distribution shift they represent: synthetic datasets, cross-domain datasets, and a temporal evolution dataset. These correspond to artificial shifts, cross-domain shifts, and temporal shifts, respectively, as summarized in Table 2. The subsequent subsections provide a detailed account of the pre-processing steps and data partitioning strategies applied to each category.

Table 2: The detailed information of the original datasets.

| Distribution shift | Dataset | #Nodes | #Edges | #Features | #Classes | Train/Val/Test |
|---|---|---|---|---|---|---|
| Artificial Shift | Cora | 2,708 | 5,429 | 1,433 | 7 | Domain-level |
| | CiteSeer | 3,327 | 4,732 | 3,703 | 6 | Domain-level |
| | PubMed | 19,717 | 44,338 | 500 | 3 | Domain-level |
| | LastFMAsia | 7,624 | 55,612 | 128 | 18 | Domain-level |
| Cross-domain Shift | Twitch | 34,120 | 892,346 | 2,545 | 2 | Domain-level |
| | WebKB | 617 | 1,138 | 1,703 | 5 | Domain-level |
| Temporal Shift | ogb-ArXiv | 169,343 | 1,116,243 | 128 | 40 | Time-level |

## D.1    SYNTHETIC DATASETS PRE-PROCESSING

Cora, CiteSeer, PubMed, and LastFMAsia are four widely utilized benchmark datasets for node classification tasks, frequently employed to evaluate the performance and design of GNNs. The Cora, CiteSeer, and PubMed datasets represent citation networks, where nodes correspond to academic papers and edges denote citation relationships between them. LastFMAsia is a social network dataset in which nodes represent users of the LastFM platform, and edges indicate friendship relations among users.

For each dataset, we first duplicate the original graph by $n_{env}$ times (the number of latent environments), each one prepared for being augmented by the corresponding spurious environment-sensitive noise. Then, we use the adjacency matrix and the label to construct the spurious noise. Specifically, assume the adjacency matrix as $\mathbf{A}$, the original node features as $\mathbf{X}_{ori}$ and the node label as $\mathbf{y}$. Then we adopt a randomly initialized GNN (with the adjacency matrix $\mathbf{A}$ and the node label $\mathbf{y}$) to generate the invariant node features, denoted as $\mathbf{X}_{inv}$. Then, we employ another randomly initialized MLP (with input of a Gaussian noise whose mean value is the corresponding environment id within $n_{env}$) to generate spurious node features $\mathbf{X}_{spu}$. By integrating the invariant and spurious node features together, we obtain the node features with the artificial distribution shift $\mathbf{X}_{art} = \mathbf{X}_{inv} + \mathbf{X}_{spu}$. After that, we concatenate the original node features and the features with artificial distribution shift $\mathbf{X} = [\mathbf{X}_{ori}, \mathbf{X}_{art}]$ as input node features for training and evaluation. In this way, we construct $n_{env} = 5$ graphs with different environment id's for each dataset. For all baselines, we use three environments for training, one for validation and report the classification accuracy on the remaining graph.

## D.2    CROSS-DOMAIN DATASETS PRE-PROCESSING

A common scenario in which distribution shifts arise in graph-structured data is cross-domain transfer. In many real-world applications, multiple observed graphs may be available, each originating from a distinct domain. For instance, in the context of social networks, domains can be defined based on the geographic or demographic context in which the networks are collected. More generally, graph data typically captures relational structures among a specific set of entities, and the nature

of interactions or relationships often varies significantly across different groups. As a result, the underlying data-generating distributions differ between domains, giving rise to domain shifts.

The Twitch and WebKB datasets exemplify cross-domain distribution shifts as described above. The Twitch dataset comprises six distinct networks, each representing a different geographical region—specifically, DE, PT, RU, ES, FR, and EN. In these networks, nodes correspond to Twitch users of game streaming, and edges represent friendship relations among them. While these networks share invariant characteristics—such as the majority of users being game streamers—they also exhibit region-specific spurious correlations (e.g., users from a particular region may demonstrate preferences for certain games). This combination of shared and domain-specific features makes Twitch a suitable dataset for evaluating OOD generalization under cross-domain shifts. For all baseline models, we use the networks from DE and PT as the training domains, those from RU and ES for validation, and the remaining networks (FR and EN) for testing.

Another dataset is WebKB which consists of three networks (i.e., Wisconsin, Cornell and Texas) of web pages collected from computer science departments of different universities. In each network, nodes represent web pages and edges represent the hyperlink between them. On the one hand, these networks share some invariant knowledge since they are all collected from the computer science departments; on the other hand, they also contain spurious correlations due to the fact that these departments from different universities may have different research focuses. As a result, these properties make WebKB an ideal OOD dataset with various cross-domain distribution shifts. For all baselines, we employ the network from Wisconsin for the training set, that from Cornell for the validation set and the remaining for the testing set.

### D.3 TEMPORAL EVOLUTION DATASETS PRE-PROCESSING

Another prevalent scenario for OOD generalization arises in the context of temporal graphs, which evolve dynamically over time through the addition or deletion of nodes and edges. As illustrated in Figure 1(a) of Section 1, such graphs are common in large-scale real-world applications, including social and citation networks, where the distributions of node features, topological structures, and labels often exhibit strong temporal dependencies at varying time scales. To investigate temporal distribution shifts in node classification, we employ the widely-used ogb-ArXiv dataset, which provides a benchmark setting for evaluating model performance under temporal dynamics. The ogb-ArXiv dataset consists of 169,343 nodes, each representing a computer science paper from the arXiv repository, with 128-dimensional feature vectors. It contains 1,116,243 edges that capture citation relationships between papers, and 40 distinct node labels corresponding to the subject areas of the papers. This dataset serves as a representative benchmark for OOD generalization under temporal distribution shifts, as citation behaviors naturally evolve over time. For all baseline models, papers published prior to 2017 are used for training, those published in 2018 for validation, and the remaining papers for testing.

## E DETAILED EXPERIMENTAL RESULTS AND ANALYSIS

In this section, we provide the additional experimental results and analysis illustrated in Section 4. Figure 4 is the experiment results of the performance comparison from Section 4.2. Table 3 is the experiment results of the ablation study from Section 4.3. Table 4 and Table 5 are the experiment results of the sensitivity study from Section 4.4. Figure 5 is the experiment results of the model visualization from Section 4.5.

### E.1 OVERALL PERFORMANCE COMPARISON

In this Section, we aim to answer **Q1**: How does the proposed LRW-OOD perform compared to state-of-the-art models on datasets under various distribution shifts? To this end, We conduct a comprehensive evaluation of our model against other models across various kinds of datasets, including synthetic datasets, cross-domain datasets and the temporal evolution dataset. The results are shown in Figure 3, Figure 4 and Table 1.

**Distribution Shifts on Synthetic Datasets.** We report the testing accuracy on the Cora, CiteSeer, PubMed, and LastFMAsia, each augmented with environment-specific spurious noise features of

Table 3: The ablation performance for LRW-OOD.

| Backbone | Model | Cora | CiteSeer | PubMed | LastFMAsia |
|----------|-------|------|----------|--------|------------|
| GCN | LRW-OOD | **82.63±0.64** | **73.79±1.69** | **82.71±0.13** | **73.62±0.45** |
| | w/o SM | 81.47±1.01 | 71.94±0.41 | 82.13±0.10 | 71.71±0.45 |
| | w/o REM | 81.80±0.57 | 70.18±0.72 | 82.16±0.04 | 73.47±0.34 |
| | w/o KDE_MI | 74.2±1.4 | 60.2±2.6 | 77.9±0.1 | 66.2±1.7 |
| | w/o LRW | 73.4±0.39 | 59.08±0.27 | 78.03±0.35 | 67.4±0.26 |
| GAT | LRW-OOD | **82.61±0.03** | **73.65±0.44** | **82.84±0.24** | **73.83±0.78** |
| | w/o SM | 81.38±0.84 | 71.72±0.17 | 81.86±0.10 | 72.48±0.87 |
| | w/o REM | 81.63±0.58 | 70.16±0.79 | 82.07±0.03 | 73.51±0.6 |
| | w/o KDE_MI | 76.2±0.8 | 61.4±2.3 | 79.7±0.8 | 68.5±2.9 |
| | w/o LRW | 75.1±0.33 | 59.25±1.03 | 78.25±0.99 | 71.79±1.34 |

varying dimensions, ranging from 20 to 160, as illustrated in Figure 3 and Figure 4. While all evaluated models exhibit a decline in performance as the dimensionality of the spurious features increases, the proposed LRW-OOD consistently achieves superior accuracy compared to all baseline methods, regardless of employing GCN or GAT as the backbone. These results highlight the robust graph OOD generalization capability of LRW-OOD in the presence of complex distribution shifts.

**Distribution Shifts on Cross-domain & Temporal Evolution Datasets.** As presented in Table 1, we report the test accuracy on the WebKB, Twitch, and ogb-ArXiv. The proposed LRW-OOD demonstrates consistently strong performance, significantly outperforming all baseline methods, despite the challenges of substantial domain/temporal distribution shifts within these datasets. Specifically, LRW-OOD achieves an average improvement of approximately 3.1% when using the GCN backbone and around 1.0% with the GAT backbone compared to the sub-optimal baselines. These results underscore the model's ability to effectively learn invariant representations while mitigating the influence of domain-specific and temporal-specific spurious correlations, thereby highlighting its potential for robust deployment in real-world scenarios.

## E.2 ABLATION STUDY

To answer **Q2**, we present a comprehensive ablation analysis of the contribution of each individual component within LRW-OOD across Cora, CiteSeer, PubMed, and LastFMAsia. The experiment results are summarized in Table 3. Specifically, w/o SM refers to the variant of LRW-OOD where the sufficiency maximization loss is modified by substituting the original KDE-based approach with KL-divergence, following established methodologies in prior literature (Wu et al. (2022); Wang et al. (2025)). w/o REM indicates the model configuration without the risk extrapolation minimization loss. w/o KDE_MI means that the model is trained by tratidional cross-entropy loss instead of the proposed KDE-baesd MI loss. Finally, w/o LRW corresponds to the vanilla random-walk-based GNN model without the proposed LRW encoder. From Table 3, it can be observed that LRW-OOD without the LRW encoder achieves the worst performance compared to that without SM loss and that without REM loss, which emphasis the great importance of the LRW encoder as the extractor of the invariant knowledge. Also, the model trained without the proposed KDE-based MI loss achieves the second worst performance, indicating the important role of the proposed loss (composed of the SM loss and REM loss altogether) when facing various distribution shifts. Meanwhile, we also observe that LRW-OOD without REM loss achieves a better performance compared to that without SM loss on most of the datasets, which demonstrates that the SM loss plays a more important role than the REM loss for the graph OOD generalization.

## E.3 SENSITIVITY ANALYSIS

To answer **Q3**, we investigate the impact of key hyperparameters—specifically, the number of random walk steps and the number of random walk times—on the performance of the proposed LRW-OOD. The evaluation is conducted across multiple datasets, including Cora, CiteSeer, PubMed,

Table 4: The performance of LRW-OOD under different hyper-paramenters of random walk steps.

| Backbone | Dataset | 1-step | 2-steps | 3-steps | 4-steps | 5-steps |
|----------|---------|--------|---------|---------|---------|---------|
| GCN | Cora | **84.4±0.11** | 83.24±0.14 | 82.66±0.37 | 82.63 ±0.64 | 82.53±0.20 |
| | CiteSeer | **74.78±0.64** | 73.58±0.43 | 73.25±0.37 | 73.79±1.69 | 72.72±0.09 |
| | PubMed | **83.14±1.32** | 82.76±0.78 | 82.8±0.43 | 82.71±0.13 | 82.53±0.54 |
| | LastFMAsia | 73.20±0.30 | 73.32±0.16 | 73.35±0.23 | 73.47±0.34 | **74.36±0.32** |
| GAT | Cora | **85.98±0.19** | 85.01±0.31 | 84.43±0.43 | 82.61±0.03 | 83.75±0.50 |
| | CiteSeer | **77.80±0.20** | 77.06±0.38 | 76.33±0.54 | 73.65±0.44 | 72.98±0.39 |
| | PubMed | **83.76±1.48** | 83.68±0.62 | 83.12±0.14 | 82.84±0.24 | 82.45±0.35 |
| | LastFMAsia | 73.44±0.37 | 73.56±0.06 | **74.18±0.14** | 73.83±0.78 | 72.92±0.15 |

Table 5: The performance of LRW-OOD under different hyper-paramenters of random walk times.

| Backbone | Dataset | 1-time | 2-times | 3-times | 4-times | 5-times |
|----------|---------|--------|---------|---------|---------|---------|
| GCN | Cora | 81.52±0.24 | 82.92±0.13 | 82.45±0.23 | 82.63±0.64 | **83.01±0.23** |
| | CiteSeer | 71.54±1.23 | 72.98±0.29 | 73.25±0.37 | **73.79±1.69** | 73.08±0.24 |
| | PubMed | 81.94±0.41 | 81.88±1.01 | 82.03±0.14 | 82.71±0.13 | **83.01±0.48** |
| | LastFMAsia | 71.90±0.54 | 72.59±0.2 | 72.43±0.20 | 73.47±0.34 | **73.96±0.13** |
| GAT | Cora | 81.90±0.43 | 82.93±0.52 | 81.49±0.28 | 82.61±0.03 | **83.62±0.43** |
| | CiteSeer | 72.94±0.67 | 73.12±0.35 | 73.33±0.54 | **73.65±0.44** | 73.22±0.42 |
| | PubMed | 81.96±0.94 | 82.08±1.14 | 82.53±0.89 | 82.84±0.24 | **83.01±0.84** |
| | LastFMAsia | 72.81±0.45 | 73.00±0.37 | 73.07±0.11 | 73.83±0.78 | **73.98±0.23** |

and LastFMAsia. The corresponding experimental results are presented in Table 4 and Table 5. As illustrated in Table 4, LRW-OOD achieves optimal performance with a single-step on the Cora, CiteSeer, and PubMed. In contrast, more walk steps are required to attain the best results on LastFMAsia. This discrepancy can be attributed to differing levels of homophily across datasets: Cora, CiteSeer, and PubMed exhibit more homophily, wherein the invariant knowledge is predominantly localized within the 1-hop neighborhood. Conversely, LastFMAsia demonstrates lower homophily, implying that invariant patterns reside in higher-order neighborhoods. As shown in Table 5, LRW-OOD consistently achieves optimal performance when multiple random walks are initiated per node across all evaluated datasets. This observation suggests that each individual random walk may capture environment-specific spurious correlations. By conducting multiple random walks, LRW-OOD is better equipped to retain invariant patterns while effectively mitigating the influence of spurious correlations across walks.

### E.4   MODEL VISUALIZATION

To address **Q4**, we visualize the LRW embeddings derived from different orders of random walk paths on four synthetic datasets: Cora, CiteSeer, PubMed, and LastFMAsia, as shown in Figure 5. The visualizations reveal that LRW embeddings generated from different walk orders exhibit clear distinctions across all datasets. Notably, these datasets are constructed with varying environment-specific spurious correlations, as described in Section 4.1. Consequently, the model learns to encode distinct representations from random walks, effectively mitigating spurious correlations and capturing invariant information shared across environments with various distribution shifts. This results in a more expressive representation that facilitates the learning of predictive relationships beneficial for graph OOD generalization.

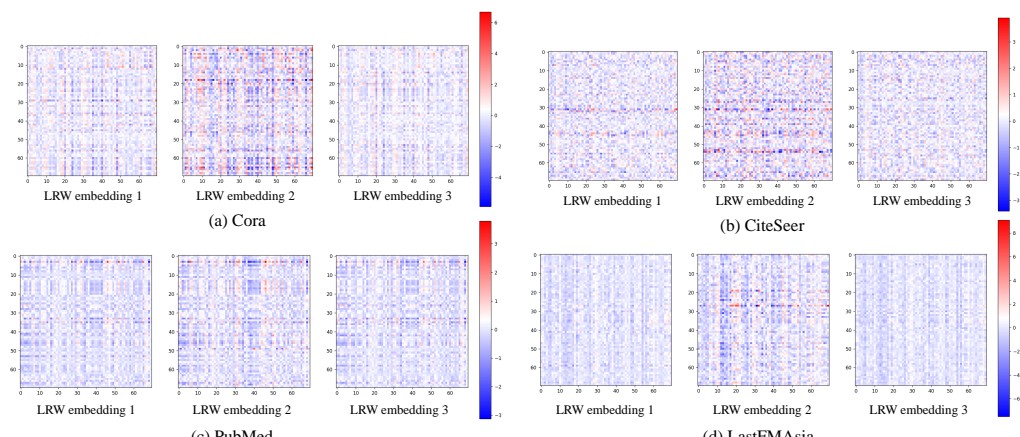

Figure 5: The visualization of weights of LRW-OOD on the synthetic datasets.

## F  LIMITATIONS AND FUTURE DIRECTIONS

In this section, we mainly examine the limitations of the proposed model and outline potential avenues for enhancing its performance and applicability. The primary drawbacks of the model are twofold: computational complexity and the hyperparameter tuning overhead.

**Existing Limitations.** (1) Computational Complexity: As stated in Theorem 3.4, the time complexity of the proposed model is $\mathcal{O}(nd^2(l_1 + l_2) + nksd)$. Since the complexity scales linearly with the number of nodes $n$ and quadratically with the feature dimension $d$, the computational cost becomes substantial in large-scale scenarios—for instance, when the graph contains over $10^9$ and the feature dimension exceeds $10^3$. (2) Hyperparameter Tuning Overhead: The hyperparameters of the proposed model are primarily tuned manually based on empirical knowledge, rather than through automated optimization techniques. This manual tuning process is both time-consuming and suboptimal in terms of efficiency and effectiveness.

Based on the aforementioned limitations, several promising research directions can be pursued to address these challenges, as outlined below.

**Future Directions.** (1) Down-sampling & Graph Condensation Strategies: For extremely large-scale graphs, it is beneficial to apply down-sampling or graph condensation techniques to reduce the graph size while preserving key topological and semantic properties. These approaches can effectively mitigate the computational burden by generating compact graph representations that approximate the original structure and information content. (2) Automated Hyperparameter Tuning Methods: Manual hyperparameter tuning is often labor-intensive and may lead to suboptimal results. To improve both the efficiency and effectiveness of this process, automated hyperparameter optimization methods—such as Optuna (Akiba et al. (2019)) and Ray Tune (Liaw et al. (2018))—can be integrated into the model training pipeline. These frameworks facilitate systematic exploration of the hyperparameter space and can significantly enhance model performance with reduced human intervention. (3) Extension to Broader Downstream Tasks: While the current work primarily evaluates the model on the node classification task, the proposed framework can be readily extended to other downstream applications such as link prediction and graph classification. For instance, in the link prediction scenario, the mutual information term $\mathcal{L}_{\text{MI\_Suff}}(\mathbf{h}_i^r, \mathbf{y}_i)$ can be replaced by $\mathcal{L}_{\text{MI\_Suff}}(\mathbf{h}_i, \mathbf{h}_j)$ and apply the KDE-based MI loss formulated as Eq. (5), where $\mathbf{h}_i, \mathbf{h}_j$ can be the aggregated LRW embeddings of their corresponding $k$ sampled random walk paths of node $v_i$ and its neighbor $v_j$, respectively. Similarly, for the graph classification task, a graph-level LRW embedding $\mathbf{h}_i$ can be obtained by aggregating the LRW embeddings of all nodes in graph $G_i$, and the mutual information term in Eq. (5) can be reformulated as $\mathcal{L}_{\text{MI\_Suff}}(\mathbf{h}_i, \mathbf{y}_i)$.

# G GenAI Usage Disclosure

In the preparation of this manuscript, we have utilized generative artificial intelligence (GenAI) tools, specifically GPT-4o and claude, to assist with text polishing and refinement, as well as to support the drafting and modification of code snippets. These tools have been employed to enhance the clarity and readability of the narrative and to facilitate the development of auxiliary code, ensuring a streamlined presentation of our work. However, we emphasize that GenAI was not utilized in the derivation of mathematical formulas, the design or implementation of key algorithms, or the formulation of core scientific insights. All critical theoretical proofs, algorithmic developments, and experimental validations were conducted independently by the authors to maintain the integrity and originality of the research. We have rigorously reviewed and verified all generated text to ensure accuracy and alignment with the scientific content, thereby upholding the reliability of the presented results.

