# OpenReview forum: "Rethinking Graph Out-Of-Distribution Generalization: A Learnable Random Walk Perspective"
_ICLR.cc/2026/Conference — Submitted to ICLR 2026_

### Official Review · Reviewer_aA5X · 2025-10-28

**Soundness:** 2
**Presentation:** 1
**Contribution:** 2
**Rating:** 4
**Confidence:** 5

**Summary:**

The authors propose LRW-OOD, a framework that replaces the conventional invariant topology or spectrum assumptions with a learnable random walk (LRW) representation to capture invariant knowledge across environments. The LRW encoder parameterizes the random walk transition matrix using a neural sampler and a path encoder, and optimizes it through a kernel density estimation (KDE)-based mutual information (MI) loss, which jointly enforces sufficiency and invariance conditions for OOD robustness.

**Strengths:**

1. LRW-OOD introduces a learnable transition matrix to address the limitations of existing topology- or spectrum-invariant assumptions.
2. The paper provides theoretical guarantees supporting the proposed method.
3. The experimental setup is extensive, covering synthetic, cross-domain, and temporal distribution shifts.

**Weaknesses:**

1. The baselines include classic models but omit very recent methods. Without these comparisons, the reported 3.87% improvement may not fully reflect the current state-of-the-art competitiveness.
2. The theoretical sections contain many notations and are difficult to follow without intuitive explanations. Several variables (e.g., m, S, u) are only implicitly defined.
3. Although Theorem 3.4 provides asymptotic complexity analysis, no empirical runtime or memory benchmarks are reported. It remains unclear whether the KDE-based MI computation scales efficiently to large graphs such as ogb-ArXiv.
4. The experimental discussion is relatively brief and lacks deeper interpretation or analysis of the results.
5. Figure 1 does not reflect the advantages of random-walk-based methods over existing Topology- and Spectrum-based methods.

**Questions:**

1. How sensitive is the KDE-based MI loss to the choice of kernel bandwidth and kernel type? Were these parameters tuned or fixed across datasets?
2. Have the sufficiency and invariance conditions discussed in Theorem 3.1 been empirically verified, for example by tracking mutual information metrics during training?
3. Are there optimization conflicts between MI sufficiency maximization and risk extrapolation minimization, and if so, how are they balanced during training?

---

> ### Author Response · Authors · 2025-11-17
>
> We sincerely appreciate your thoughtful and constructive feedback, as well as your recognition of our theoretical framework and the experimental evaluation. In the following, we will address the weaknesses and questions you highlighted in your review.
>
> [W1] Thank you very much for raising this issue. We include two additional recent baselines (i.e., GRM and DNRL) into the performance comparison, which are all released in 2025. Please review the revised Figure 3, Figure 4 and Table 1 in Section 4.2.
>
> [W2] Thank you very much for raising this question. We apologize for not sufficiently illustrating the definition of several notations in our paper. We have increased the intuitive explanations about these notations. Please review the revised blue-highlighted content in Section 3.2.
>
> [W3] Thank you very much for raising this question. In graph OOD generalization, existing researches still mainly focus on improving the OOD performance under various distribution shifts instead of optimizing the time and space complexity of models. You can see some of the baselines (i.e., SpCo, CiT, MARIO and DNRL) even requires more than 80GB GPU memory when tested on ogb-ArXiv in Table 1, despite the fact that they indeed achieve relatively well OOD performance on other datasets.
>
> As for our models, the time complexity of the proposed model scales linearly with the number of nodes n and quadratically with the feature dimension d, which means it will become relatively large when the number of graph nodes is over $10^5$ nodes and the feature dimension exceeds $10^2$, as the case of ogb-ArXiv, as illustrated in Theorem 3.4 and in Appendix F. However, we still achieve the SOTA performance over other baselines on ogb-ArXiv. Also, in order to optimize the time and space complexity, we plan to take multiple actions, such as down-sampling and graph condensation strategies to reduce the computation requirement in the future, as illustrated in Appendix F.
>
> [W4] Thank you very much for raising this issue. Unfortunately, due to the page limit, we have to put the simplified analysis of the experiment results in the main text. However, we put the detailed analysis in Appendix E. We kindly invite you to review the relevant content.
>
> [W5] Thank you for raising this question. The function of Figure 1 is to give an overview about the difference between the existing topology & spectrum-based models and the random-walk-based models, as well as illustrating the example of heterophilic citation networks under temporal distribution shifts altogether. Unfortunately, due to the page limit, we can only put one figure in Section 1. Thus, we choose to illustrate the advantages of random-walk-based models in the form of words (i.e., L1, L2, M1 and M2 in Section 1) and images (i.e., Figure 1(a)) together.
>
> [Q1] Thank you very much for raising this question. The primary reason we use KDE is to estimate the latent and unknown label distribution $\mathrm{Pr}(\textbf{y})$ while maintaining the differentiability of the loss. Since KDE itself is not our major contribution, we do not conduct sensitivity experiments on the choice of kernel bandwidth and kernel type. Instead, we follow the common way of existing KDE works, using a Gaussian kernel with a fixed bandwidth of 1, as illustrated in Equation (4). And these hyper-parameters are fixed during our experiments.
>
> [Q2] Thank you very much for raising this question. Yes, the sufficiency and invariance conditions are empirically validated. The KDE-based MI loss is essentially the negative form of mutual information, and it is minimized during training. This optimization process encourages the model to satisfy the sufficiency and invariance conditions. To further support this, we have included additional ablation experiments in Table 3, comparing the performance of models trained with and without the KDE-based MI loss. The results demonstrate a substantial performance degradation when the proposed loss is omitted, providing empirical evidence for the effectiveness of both conditions.
>
> [Q3] Thank you for raising this question. We don’t think there is any optimization conflicts between MI sufficiency maximization and risk extrapolation minimization. According to Theorem 3.1 and its proof in Appendix C.1, achieving MI sufficiency maximization requires maximizing the mutual information between h and y, and achieving risk extrapolation minimization needs minimizing the variance among $ \\{ \mathcal{L}_ {\mathrm{MI-Suff}}(\textbf{h}_ i^r,\textbf{y}_ i) \\}_ {r=1}^k$. There isn’t any theoretical conflict between these two requirements. For example, if all k sampled negative mutual information $\mathcal{L}_ {\mathrm{MI-Suff}}(\textbf{h}_ i^r,\textbf{y}_ i)$ are low enough (e.g., they are all equal to -1), this means the mutual information between h and y is high, while the variance of these mutual information is very low. Thus, both MI sufficiency maximization and risk extrapolation minimization are achieved in this case.

---

> ### Author Response · Authors · 2025-11-27
>
> Dear Reviewer aA5X,
>
> We hope this message finds you well. We have updated our rebuttal and our paper according to your reviews. As the discussion time is nearing to its end, we want to know if we address the concerns satisfactorily. If there are any additional points or feedbacks you'd like us to consider, please let us know. Your insights are invaluable to us, and we're eager to to address any remaining issue to improve our work.
>
> Thank you for your time and effort.
>
> Best regards,
>
> The authors of Paper 12627

---

### Official Review · Reviewer_Ai73 · 2025-10-30

**Soundness:** 3
**Presentation:** 2
**Contribution:** 3
**Rating:** 6
**Confidence:** 3

**Summary:**

This paper introduces LRW-OOD, a model for graph Out-Of-Distribution (OOD) generalization. It replaces traditional fixed transition matrices with a learnable random walk (LRW) sampler and path encoder to capture invariant knowledge across distribution shifts. The model uses a KDE-based mutual information loss to generate random walk sequences aligned with OOD principles. The paper theoretically demonstrates the connection between random walk sequences and graph OOD generalization, achieving an average improvement of 3.87% in experimental performance.

**Strengths:**

Originality: The idea of using learnable random walk (LRW) sequences to capture invariant features across distribution shifts is innovative and interesting, offering a fresh perspective on graph OOD generalization.

Theoretical Completeness: The paper provides solid theoretical grounding, demonstrating through rigorous proofs that LRW sequences can effectively capture invariant knowledge and align with OOD principles.

Experimental Validation: Extensive experiments on Synthetic Datasets, Cross-domain, and Temporal Evolution Datasets show that the proposed LRW-OOD model is effective in various OOD scenarios, outperforming existing methods.

**Weaknesses:**

Clarity and Presentation: Some sections could benefit from additional clarification. For example, the introduction's example is somewhat unclear and could be more straightforward. Additionally, the experimental setup on different datasets, while detailed in the appendix, would be clearer if briefly introduced in the main text.

Insufficient Explanation of Random Walk Approach: The paper uses random walk sequences to address the limitations of existing methods, but more detailed explanation or case studies are needed to better clarify how this approach effectively addresses the challenges and to enhance understanding of LRW-OOD's underlying mechanisms.

Limited Task Generalization: The paper primarily evaluates the model on node classification tasks. Further exploration of its performance on other graph-based tasks, such as graph classification, would provide a better understanding of the model's general applicability.

**Questions:**

Aside from the weaknesses mentioned, a technical question arises: For single-variable probability prediction, KDE can be used as shown in the right-hand side of Equation (4). How is the joint probability distribution for two variables specifically modeled in this case?

Some notations in the methodology section need further clarification. For example, in Equation (5), does V represent variance?

---

> ### Author Response · Authors · 2025-11-17
>
> We sincerely appreciate your thoughtful and constructive feedback, as well as your recognition of our research idea, theoretical framework and the experimental evaluation. In the following, we will address the weaknesses and questions you highlighted in your review.
>
> [W1] Thank you very much for raising these issues. We apologize for not sufficiently illustrating the introduction’s example, which may cause your confusion. The example illustrates a heterophilic citation network under temporal distribution shift, where both graphs share similar topological while different semantic information. In this case: (1) if subgraphs are used as carriers of invariant knowledge, the model may fail to accurately extract such invariant subgraphs with similar topological structures but different semantics; (2) if graph spectra are used as carriers of invariant knowledge, the model may fail to accurately extract invariant spectra among graphs with strong homophily and those with strong heterophily due to the underlying temporal distribution shift. Moreover, we make a more detailed case study in Appendix B to illustrate the example. We kindly invite you to review it.
>
> Meanwhile, as for the illustration for datasets, we follow your advice and include more detailed introduction in the main text. Please review the revised blue-highlighted content in Section 4.1.
>
> [W2] Thank you very much for raising this issue. We apologize for not sufficiently illustrating the mechanism of proposed learnable random walk sequences to solve existing models’ limitations. To illustrate, taking Figure 1 as an example, during the early stages of training, the LRW sampler performs random walks based on initial feature similarity, which tends to aggregate information from homophilic neighbors. In the case of the 1990s citation network, physical nodes tend to aggregate information from other physical nodes, often ignoring mathematical nodes. However, as the training process continues, the KDE-based MI loss function, based on OOD principles, helps the LRW sampler identify spurious correlations in the initial probability transition matrix, which was based on feature similarity. As a result, the LRW sampler dynamically adjusts the features of each node to retain the invariant information and discard the spurious correlations. This allows the model to aggregate information from math nodes in the 2020s citation network, enabling correct node classification tasks in the 2020s networks.
>
> Moreover, we have further validated this explanation in Section 4.2 with experiments on the ogb-ArXiv dataset. In our experiments, we used citation networks prior to 2017 as the training set, the 2018 network as the validation set, and the remaining networks as the test set. The results show that our model accurately extracts invariant patterns in temporal distribution shifts while discarding spurious correlations, leading to excellent performance.
>
> We have included the detailed explanation about the mechanism of our proposed method in Appendix B. Please review the revised blue-highlighted content.
>
> [W3] Thank you for raising this issue. It’s true that we primarily focus on node classification tasks in this paper. However, the proposed methodology can be extended into broader downstream tasks, such as link prediction and graph classification, as illustrated in Appendix F. For instance, in the link prediction scenario, the mutual information term $\mathcal{L}_ {\operatorname{MI-Suff}}(\textbf{h}_ i^r,\textbf{y}_ i)$ can be replaced by $\mathcal{L}_ {\operatorname{MI-Suff}}(\textbf{h}_ i,\textbf{h}_ j)$ and apply the KDE-based MI loss formulated as Eq. (5), where $\textbf{h}_ i, \textbf{h}_ j$ can be the aggregated LRW embeddings of their corresponding $k$ sampled random walk paths of node $v_i$ and its neighbor $v_j$, respectively. Similarly, for the graph classification task, a graph-level LRW embedding $\textbf{h}_ i$ can be obtained by aggregating the LRW embeddings of all nodes in graph $G_ i$, and the mutual information term in Eq. (5) can be reformulated as $\mathcal{L}_ {\operatorname{MI-Suff}}(\textbf{h}_ i,\textbf{y}_ i)$. We will carry out the relevant research in the future.
>
> [Q1] Thank you very much for raising these questions. We apologize for not sufficiently illustrating the detailed process of calculating the probability distribution for $\textbf{h}_ i^r$ and $\textbf{y}_ i$. As illustrated in Equation (4), we choose Gaussian kernel to estimate the distributions, which is in essence the mathematical formula of normal distribution. Normal distribution has the version of joint probability distribution, and we simply employ this.
>
> [Q2] Thank you for raising this question. Yes, we follow the way of normal notation representation from probability and $\mathbb{V}$ represents the variance. We are sorry for not clarifying these notations more clearly, and we have revised the relevant content. Please review the blue-highlighted part in Section 3.2.

---

> ### Comment · Reviewer_Ai73 · 2025-11-26
>
> I would like to thank authors for their detailed responses. I’m pleased to see the detailed explanation of the introduction example added to the appendix, which is essential to improve the paper’s motivation and presentation. The revised manuscript has addressed my concerns and I will maintain my positive score.

---

> > ### Author Response · Authors · 2025-11-26
> >
> > Thank you for your response and positive assessment. We again sincerely appreciate your thoughtful feedback and are glad to hear that the revisions have addressed your concerns. Your insights are invaluable in improving the clarity and motivation of our work.

---

### Official Review · Reviewer_Wcg9 · 2025-11-01

**Soundness:** 3
**Presentation:** 3
**Contribution:** 2
**Rating:** 4
**Confidence:** 3

**Summary:**

This paper tackles the problem of out-of-distribution OOD generalization for graph neural networks, a challenge due to distribution shifts in topology or node features. The authors propose LRW-OOD that reinterprets invariant knowledge as learnable random walk sequences rather than relying on invariant topology or spectral properties. The framework combines an LRW-sampler, a path encoder, and a kernel density estimation-based mutual information loss to enhance invariance and reduce spurious correlations. Experiments on seven benchmark datasets show improvements of about 3–4% over existing baselines under various distribution shifts.

**Strengths:**

The paper addresses an important and timely problem in graph learning OOD generalization under distribution shifts. The proposed idea of viewing invariance through learnable random walks provides an interesting alternative to existing topology- and spectrum-based paradigms. The method is clearly described, with mathematical formulations and theoretical discussions that enhance readability. Experimental evaluations are good, covering multiple datasets and baseline comparisons.

**Weaknesses:**

The paper’s conceptual novelty is somewhat limited. The learnable random walk formulation largely reformulates known ideas about adaptive sampling and mutual-information-based regularization, without a fundamentally new theoretical insight. The related work section lacks a full discussion of prior random-walk-based or path-level GNN models, making it unclear how LRW-OOD truly departs from those approaches. Moreover, the experimental baselines, while broad, omit several recent and strong graph OOD methods that incorporate causal intervention or environment simulation. The result analysis is descriptive; the claimed theoretical benefits are not convincingly validated through ablation or controlled tests linking the KDE-based MI loss to invariance improvement.

**Questions:**

Could the authors elaborate on how the proposed learnable transition matrix differs in effect from existing adaptive adjacency mechanisms? How does the model behave when applied to large-scale graphs where random walk sampling becomes computationally expensive? Thanks.

---

> ### Author Response · Authors · 2025-11-17
>
> We sincerely appreciate your thoughtful and constructive feedback, as well as your recognition of our research idea, theoretical framework and the experimental evaluation. In the following, we will address the weaknesses and questions you highlighted in your review. Due to the characters limit, we have to simplify your questions and answer them point by point.
>
> [W1] Conceptutal Novelty Limitation: Thank you very much for raising this question. As stated in Section 1, the main contributions of our work are (1) introducing a new perspective on learnable random walk sequences for graph OOD generalization, and (2) proposing a new learning paradigm based on this perspective. These contributions go beyond adaptive sampling or mutual-information-based regularization. We believe that providing a new viewpoint, even using existing concepts, can be as valuable to machine learning as introducing new concepts.
>
> A well-known example is the PPO algorithm in reinforcement learning. While PPO uses classical elements—such as the policy-critic-reward architecture, KL-divergence penalties, and a clipping operator—its significance comes from the novel insight these ideas are combined. This new perspective produced a highly influential algorithm, which plays a key role in the development of ChatGPT and has a major impact on modern machine learning.
>
> In a similar spirit, our work aims to contribute a fresh viewpoint and a novel graph OOD learning paradigm via learnable random walk, thereby advancing the understanding of graph OOD generalization.
>
> [W2] Limited Related Work Discussion: Thank you for raising this question. To the best of our knowledge, existing graph OOD studies have not yet approached the problem from the perspective of random walks or path-level GNNs. However, the limitations of current topology-based and spectrum-based models, as discussed in Section 1, were identified through our systematic review of prior work. During this process, we observed that a random-walk-based paradigm has the potential to address these limitations, as random walks naturally integrate both topological and semantic information. Motivated by this insight, we revisited the graph OOD problem from a random-walk perspective, which ultimately led to the development of this paper. We hope that our work will encourage further research on random-walk-based or path-based approaches to graph OOD generalization.
>
> [W3] More Recent Work Comparisons: Thank you very much for raising this question. We include two additional recent baselines (i.e., GRM and DNRL) into the performance comparison, which are all released in 2025. We kindly invite you to review the revised Figure 3, Figure 4 and Table 1 in Section 4.2.
>
> [W4]: Limited Analysis & Ablation Study: A: Thank you very much for raising these issues. Unfortunately, due to the page limit, we have to simplify the result analysis in the main text. To address your concern, we put the detailed analysis in Appendix E. We kindly invite you to review it. Meanwhile, in order to validate the performance of the proposed KDE-based MI loss, we follow your suggestion and supplement the ablation study of the model without KDE-based MI loss. Please review the revised Table 3 in Section 4.3.
>
> [Q] Thank you for raising this question. Existing adaptive adjacency models still suffer from the limitations illustrated in Section 1. Taking Figure 1 for example, these models fail to extract the invariant knowledge by adaptively adjusting the adjacency matrix since these two citation networks share exactly same topology but only different semantic information (i.e., node features and labels).
>
> However, our proposed model addresses this dilemma through the dynamic behavior of the LRW sampler. Early in training, the sampler follows initial feature similarities, often aggregating information from homophilic neighbors—e.g., physical nodes in the 1990s citation network primarily gather from other physical nodes. As training progresses, the KDE-based MI loss identifies spurious correlations in the initial transition matrix and guides the sampler to retain invariant information while discarding these correlations. This enables the model to aggregate relevant information from math nodes in the 2020s network, supporting correct node classification.
>
> As for the model's behavior under computationally expensive scenarios, the OOD performance of the proposed model is theoretically supported by Theorems 3.1, 3.2, and 3.3, and is further validated empirically on the large-scale ogb-ArXiv dataset, as reported in Table 1.
>
> Regarding computational complexity, the proposed model may become relatively demanding when both the number of nodes and the feature dimension are extremely large, as discussed in Theorem 3.4 and Appendix F. To mitigate this, we plan to explore several strategies to optimize runtime and memory demands, including down-sampling and graph condensation techniques, as outlined in Appendix F.

---

> ### Author Response · Authors · 2025-11-27
>
> Dear Reviewer Wcg9,
>
> We hope this message finds you well. We have updated our rebuttal and our paper according to your reviews. As the discussion time is nearing to its end, we want to know if we address the concerns satisfactorily. If there are any additional points or feedbacks you'd like us to consider, please let us know. Your insights are invaluable to us, and we're eager to to address any remaining issue to improve our work.
>
> Thank you for your time and effort.
>
> Best regards,
>
> The authors of Paper 12627

---

### Official Review · Reviewer_KVvV · 2025-11-03

**Soundness:** 3
**Presentation:** 3
**Contribution:** 3
**Rating:** 6
**Confidence:** 3

**Summary:**

This paper tackles the critical challenge of out-of-distribution generalization for graph neural networks on node-level tasks. To overcome the limitations of existing methods, such as invariant topology or spectrum, the authors propose a novel framework called LRW-OOD. This framework is designed to extract invariant topology and knowledge patterns, enabling improved OOD generalization for node-level predictions.

**Strengths:**

1. The shift from seeking invariant topology/spectrum to learning invariant transition probabilities via random walks is interesting.

2. The paper is theoretically rigorous: rather than relying on heuristics, it formally links the learnable random walk framework to the OOD generalization objective.

3. The evaluation is thorough, covering multiple types of distribution shifts and a wide array of nine competitive baselines

**Weaknesses:**

1. While the proposed random walk method effectively extracts invariant knowledge through feature similarity, it appears less capable of capturing invariant subgraphs. This limitation may reduce its effectiveness under strong topological distribution shifts.

2. The ablation study on LRW uses a vanilla random walk GNN. A more informative comparison would include a variant employing a non-random-walk method, which could more clearly highlight the specific contribution of LRW.

3. The paper mentions a "two-stage training paradigm", but the description in Section 3.2 and Algorithm 1 makes it seem like an end-to-end process where the LRW encoder and the GNN classifier are trained jointly. Authors are expected to clarify it.

**Questions:**

see weakness

---

> ### Author Response · Authors · 2025-11-17
>
> We sincerely appreciate your thoughtful and constructive feedback, as well as your recognition of our research motivation, theoretical framework and the experimental evaluation. In the following, we will address the weaknesses you highlighted in your review.
>
> [W1] Thank you very much for raising this question. We apologize for not sufficiently illustrating the strength of learnable random walk capturing invariant subgraphs, which may cause the misunderstanding. As illustrated in Section 1 and Section 3.1, the proposed learnable random walk is capable of capturing invariant patterns across distribution shifts, which includes strong topological distribution shifts. This is due to the nature of random walk itself. If there exist invariant subgraphs under strong topological distribution shifts, the probability of walking into similar neighbors remains generally unchanged. Thus, it is capable of capturing invariant subgraphs under such distribution shifts.
>
> [W2] Thank you very much for raising this question. We apologize for not sufficiently illustrating the related terms, which may cause the misunderstanding. We have included the performance comparison using non-random-walk method in Figure 3, Figure 4 and Table 1 of Section 4.2, flagged as ERM (i.e., Empirical Risk Minimization for short). In OOD learning, ERM usually stands for the naïve method using only cross-entropy loss. Thus, in our experiment setting, the results of ERM using GCN and GAT as backbones are those employing classical GCN and GAT architectures, which are non-random-walk methods.
>
> Meanwhile, in order to validate the performance of the proposed KDE-based MI loss, we also supplement the ablation study of the model without KDE-based MI loss. Please review the revised Table 3 in Section 4.3.
>
> [W3] Thank you very much for raising this question, and we apologize for not providing a sufficiently detailed explanation of the GNN classifier training process in the original submission, which may have led to some misunderstanding. As stated in the first paragraph of Section 3.2, the main objective of our work is to facilitate graph OOD generalization through learnable random walks. For this reason, we kept the description of the GNN classifier concise. The classifier itself follows standard architectures—GCN and GAT—which serve as the backbone models in Figure 3, Figure 4, and Table 1 of Section 4.2, and is trained using the conventional cross-entropy loss.
>
> To address your concern, we have added the necessary details to Section 3.2. We kindly invite you to review the revised content, which is highlighted in blue.

---

> ### Author Response · Authors · 2025-11-27
>
> Dear Reviewer KVvV,
>
> We hope this message finds you well. We have updated our rebuttal and our paper according to your reviews. As the discussion time is nearing to its end, we want to know if we address the concerns satisfactorily. If there are any additional points or feedbacks you'd like us to consider, please let us know. Your insights are invaluable to us, and we're eager to to address any remaining issue to improve our work.
>
> Thank you for your time and effort.
>
> Best regards,
>
> The authors of Paper 12627

---

### Author Response · Authors · 2025-11-25

Dear reviewers,

We hope this message finds you well. We have updated our rebuttal and our paper according to your reviews, and we want to know if we address the concerns satisfactorily. If there are any additional points or feedbacks you'd like us to consider, please let us know. Your insights are invaluable to us, and we're eager to to address any remaining issue to improve our work.

Thank you for your time and effort.

Best regards,

The authors of Paper 12627

---

### Author Response · Authors · 2025-12-02
**Summary of Paper 12627's Rebuttal**

Dear AC,

We sincerely thank you and all the reviewers for your time and valuable feedback. Due to the recent unforeseen incident, we were unable to receive reviewers’ responses to our rebuttal. For your convenience, we provide a brief summary of the current situation.

Most reviewer comments focus on clarifying our model design and improving the presentation (e.g., Reviewer KVvV: W1–W3; Reviewer Wcg9: W2, W4, and questions; Reviewer Ai73: W1–W2 and Q2; Reviewer aA5X: W2–W5). This indicates that the motivation, novelty, theoretical guarantees, and empirical results of the original paper are already generally recognized. And we have addressed these concerns in our rebuttals and revised the paper accordingly, with all changes highlighted in blue.

Some reviewers requested additional experiments, including comparisons with recent methods and ablation studies (e.g., Reviewer Wcg9: W3–W4; Reviewer aA5X: W1). We conducted these experiments and included the results in the revised manuscript.

We also carefully addressed the reviews about the technical details, such as Q1 of Reviewer Ai73, questions of Reviewer aA5X and so on, in our rebuttal.

We again sincerely appreciate your effort and consideration.

Best regards,

The authors of Paper 12627

---

### Meta-Review · Area_Chair_asu9 · 2026-01-03

**Summary:**

The paper proposes a novel perspective on graph OOD generalization using learnable random walks (LRW-OOD). While the idea was generally found to be interesting, the reviewers raised several significant concerns that, taken together, indicate the submission is not yet ready for acceptance.

The primary concerns cluster around three areas:

1.  Limited and Contested Novelty: Reviewers Wcg9 and aA5X questioned the conceptual novelty, viewing the method as a reformulation of existing ideas (adaptive sampling, MI regularization) rather than a fundamental breakthrough. The related work was criticized for insufficient discussion of prior random-walk-based approaches.
2.  Incomplete and Unconvincing Validation: Multiple reviewers (KVvV, Wcg9, aA5X) found the experimental validation lacking. Criticisms included the omission of recent strong baselines, insufficient analysis linking the proposed KDE-based MI loss to actual invariance improvement, and a lack of runtime/scalability benchmarks for large graphs. The ablation studies were deemed inadequate.
3.  Deficiencies in Clarity and Presentation: Reviewers Ai73 and aA5X highlighted issues with clarity, including an unclear introductory example, poorly explained theoretical notations, and a presentation that made the paper difficult to follow.

**Reviewer Concerns:**

*   Addressed Concerns: The rebuttal satisfactorily clarified training procedures, added requested baseline comparisons (GRM, DNRL), and provided more detailed explanations for the introductory example and some notations. Reviewer Ai73 confirmed their concerns were addressed and maintained a positive score.
*   Outstanding & Significant Concerns:
    1.  Fundamental Novelty: The rebuttal's defense of novelty by analogy to PPO is not compelling. The core criticism—that the method is an incremental combination of known components rather than a paradigm shift—stands. The related work gap regarding prior random-walk models remains inadequately addressed.
    2.  Insufficient Empirical Proof-of-Concept: The validation remains descriptive. The critical request from Reviewer Wcg9 for "controlled tests linking the KDE-based MI loss to invariance improvement" was not met. The added ablation (with/without MI loss) shows it's important, but not *how* or *why* it enforces invariance. Scalability concerns (aA5X) were acknowledged but not empirically benchmarked, with solutions deferred to future work.
    3.  Presentation and Motivation: While some clarifications were added, the overall presentation, especially in the theoretical sections, remains a barrier. The paper’s core motivating example and the advantage of the LRW perspective over alternatives are still not communicated with crystal clarity.

**Reviewer Scores:**

*   Reviewer KVvV (Initial: 6): Likely would have maintained a borderline score. Their concerns were practical (ablation, clarification) and were addressed, but the unresolved broader issues might prevent a score increase.
*   Reviewer Wcg9 (Initial: 4): Unlikely to have raised their score significantly. Their primary concern about "limited conceptual novelty" and the lack of deep analytical validation persists.
*   Reviewer Ai73 (Initial: 6): Explicitly maintained their score of 6 after the rebuttal, satisfied with the clarifications provided.
*   Reviewer aA5X (Initial: 4): Given their high confidence and specific, unmet requests (scalability benchmarks, deeper analysis, intuitive theory), it is unlikely the rebuttal would have convinced them. Their pointed questions about sensitivity and empirical verification of theoretical conditions received defensive or incomplete answers.

---

### Decision · Program_Chairs · 2026-01-26

Reject